# Influenza virus infection expands the breadth of antibody responses through IL-4 signalling in B cells

Kosuke Miyauchi[1], Yu Adachi[2], Keisuke Tonouchi[2], Taiki Yajima[3], Yasuyo Harada[3], Hidehiro Fukuyama[4], Senka Deno[5,6,7], Yoichiro Iwakura [8], Akihiko Yoshimura[9], Hideki Hasegawa[10], Katsuyuki Yugi[5,6], Shin-ichiro Fujii[11], Osamu Ohara[12], Yoshimasa Takahashi [2] & Masato Kubo [1,3✉]

Influenza viruses are a major public health problem. Vaccines are the best available countermeasure to induce effective immunity against infection with seasonal influenza viruses; however, the breadth of antibody responses in infection versus vaccination is quite different. Here, we show that nasal infection controls two sequential processes to induce neutralizing IgG antibodies recognizing the hemagglutinin (HA) of heterotypic strains. The first is viral replication in the lung, which facilitates exposure of shared epitopes that are otherwise hidden from the immune system. The second process is the germinal center (GC) response, in particular, IL-4 derived from follicular helper T cells has an essential role in the expansion of rare GC-B cells recognizing the shared epitopes. Therefore, the combination of exposure of the shared epitopes and efficient proliferation of GC-B cells is critical for generating broadly-protective antibodies. These observations provide insight into mechanisms promoting broad protection from virus infection.

[1] Laboratory for Cytokine Regulation, Research Center for Integrative Medical Sciences (IMS), RIKEN Yokohama Institute, Yokohama, Kanagawa, Japan. [2] Research Center for Drug and Vaccine Development, National Institute of Infectious Diseases, Tokyo, Japan. [3] Division of Molecular Pathology, Research Institute for Biomedical Science, Tokyo University of Science, Chiba, Japan. [4] Laboratory for Lymphocyte Differentiation, Research Center for Integrative Medical Sciences (IMS), RIKEN Yokohama Institute, Yokohama, Kanagawa, Japan. [5] Laboratory for Integrated Cellular Systems, Research Center for Integrative Medical Sciences (IMS), RIKEN Yokohama Institute, Yokohama, Kanagawa, Japan. [6] Institute for Advanced Biosciences, Keio University, Kanagawa, Fujisawa, Japan. [7] Systems Biology Program, Graduate School of Media and Governance, Keio University, Kanagawa, Fujisawa, Japan. [8] Center for Animal Disease Models, Research Institute for Biomedical Science, Tokyo University of Science, Chiba, Japan. [9] Department of Microbiology and Immunology, Keio University School of Medicine, Tokyo, Japan. [10] Influenza Virus Research Center, National Institute of Infectious Diseases, Tokyo, Japan. [11] Laboratory for Immunotherapy, Research Center for Integrative Medical Sciences (IMS), RIKEN Yokohama Institute, Yokohama, Kanagawa, Japan. [12] Laboratory for Integrative Genomics, Research Center for Integrative Medical Sciences (IMS), RIKEN Yokohama Institute, Yokohama, Kanagawa, Japan. ✉email: masato.kubo@riken.jp

nfluenza viruses are airborne pathogens that cause mild to severe respiratory infections and periodic pandemics. Viruses have caused pandemics in the past and then became seasonal infections within a few years, but they still kill over 290,000 people annually worldwide. Pandemics are typically caused by viruses to which the human immune system is relatively naïve. Vaccination is an effective protective strategy against influenza virus infection, however, influenza viruses can escape the host immune response owing to their high frequency of point mutations (antigenic drift) and antigenic flexibility. Therefore, discordance between the vaccine strains and the circulating strains is an unavoidable risk leading to low vaccine efficacy. Thus, the development of influenza vaccines that are not susceptible to antigenic discordance is an urgent necessity.

It has been shown previously that vaccination of C57BL/6J mice with inactivated A/H1N1pdm09 virus and highly pathogenic avian H5N1 virus predominantly induced GC-dependent and -independent antibody responses[1]. The adaptive immune responses induced by vaccination are quite different from those resulting from natural infection[2,3]. The inactivated influenza virus vaccines are highly effective in inducing protective antibodies against virus infection, but this response is quite narrow because the protection is only effective against the influenza strains used in the vaccine[4,5]. There is some supportive evidence that immune responses in a natural infection are relatively broad and have different immunodominant epitopes than those elicited with the inactivated vaccine[6–8].

Hemagglutinin (HA) is an important target of the neutralizing antibodies that interfere with virus entry and is immunodominant in the responses of mammals and birds[9,10]. HA is a trimer consisting of a stalk region and a globular domain containing the receptor-binding site (RBS). The inactivated vaccine predominantly induces antibodies recognizing the globular head domain of HA, and these antibodies generally correlate with the GC response[11–13]. However, these regions are highly susceptible to antigenic drift and the inactivated vaccine is sensitive to these antigenic changes.

By contrast, several recent observations indicate that natural infection provides opportunities to generate antibodies reacting with heterosubtypic influenza virus strains[2,14]. The natural infection resulted in a different immunodominance hierarchy than vaccination[2]. Indeed, vaccination rapidly induced antibodies recognizing the globular domain, while the stalk-specific Abs are very rare because of the restricted accessibility of the HA stalk domain[11,15,16]. In a mouse model, vaccination with the HA stalk itself or the increased local concentration of full-length HA induced stalk-specific Abs[17]. In human studies, broadly neutralizing Abs (bnAbs) have been cloned from memory B cells of infected individuals and these antibodies mainly target the region of the HA stalk domain, which is highly conserved among group 1 and group 2 influenza A viruses, and heavy-chain variable ($V_H$) region genes encoding these antibodies are heavily mutated[18,19]. These results strongly suggest that somatic hypermutation (SHM) of immunoglobulin (Ig) genes in the GC is critical for high-affinity binding to heterosubtypic HA antigenic determinants[13]. In contrast, the germline version of the human $V_H$ gene IGHV1-69 conferred pre-existing immunity without SHM by recognition of a bnAb epitope on the HA stalk[20]. Therefore, how GC responses in mediastinal lymph nodes (MLN) and GC-mediated SHM contribute to the bnAbs elicited by natural infection and the difference between the response to infection versus vaccination remain unanswered questions.

The GC is generated as a defined structure in a B cell follicle of secondary lymphoid organs in response to infection and immunization. Activated B cells can differentiate into short-lived antibody-secreting plasma cells or enter into a GC, where affinity maturation of the B cell receptor (BCR), clonal diversification, and

isotype switching take place[21]. The GC B cells further undergo clonal expansion triggered by cognate interactions with follicular T helper ($T_{FH}$) cells in the GC light zone. The positive selection of GC B cells is driven by a BCR signal and a CD40 signal provided by CD40L on $T_{FH}$ cells. Downstream of either BCR or CD40, PI3K-AKT, and NF-κB pathways lead to upregulation of c-Myc, which is an important transcription factor for the positive selection process[22]. It has become recognized that regulation of metabolic pathways, including the mTOR pathway, plays a critical role in cell fate and functions of GC-B cells[23]. More recently, GC-B cells were reported to largely depend on fatty acids (FAs) to conduct oxidative phosphorylation (OXPHOS)[24]. However, how $T_{FH}$ cells contribute to the clonal selection or expansion of the bnAb responses has yet to be determined.

In this work, we show that viral replication during natural infection is an important process to generate broadly protective Abs recognizing the HA antigen of heterotypic influenza virus. Production of broadly protective Abs mostly depends on the GC response in the MLN, because protection against lethal challenge with heterotypic H1N1 virus is abolished in mice with reduced numbers of GC or lacking $T_{FH}$ cells. Furthermore, we also show the importance of IL-4 signaling during the clonal expansion of GC-B cells generating broadly protective Abs.

## Results

**The infection and the vaccination control distinct antibody responses.** We have previously demonstrated that the inactivated vaccine was effective in inducing protective antibodies[1]. These antibodies recognized a 30% difference in amino acid sequence in HA antigen between the pandemic strain, A/Narita/1/2009 which is a pdm2009 virus, and the seasonal H1N1 strain, A/PR/8/1934. We first compared the protective immunity between the inactivated virus vaccination and live virus infection. Mice were treated by intraperitoneal immunization with the inactivated Narita virus or by nasal infection with the live Narita virus. Fourteen days later, the treated mice were challenged with a lethal dose of A/Narita/1/2009, seasonal A/PR/8/1934, A/Okuda/1957 (H2N2), or X31 (H3N2) virus. The nasal infection with A/Narita/1/2009 was effective against not only A/Narita/1/2009 that was used for initial priming but also against A/PR/8/1934. Conversely, A/PR/8/1934 infection-induced adequate protection against A/Narita/1/2009 (Fig. 1a and Supplementary Fig. 1a). In contrast, nasal infection with A/Narita/1/2009 failed to protect against H2N2, Okuda, or H3N2, X31 viruses. These results indicated that protective immunity induced by initial priming with live A/Narita/1/2009 extended the breadth of protection to include heterotypic H1N1 influenza virus strains.

Next, we investigated whether this broad protection was due to antibody or CTL responses. Thus, we obtained bronchoalveolar lavage fluid (BALF) and serum from mice 14 days after infection with A/Narita/1/2009, and the BALF or sera were intranasally transferred into unimmunized C57BL/6J hosts, which were then infected with a lethal dose of A/Narita/1/2009 or A/PR/8/1934 to evaluate protection. BALF and serum from the mice nasally infected with live virus contained high levels of IgG and IgA antibodies specific for the Narita-HA and were equally protective against a lethal dose of A/Narita/1/2009 and A/PR/8/1934 (Fig. 1b, c). Meanwhile, serum from the vaccinated mice showed no protection against A/PR/8/1934 even though these sera contained equal anti-Narita-HA IgG levels (Supplementary Fig. 1b).

To establish which antibody class among IgA, IgM, and IgG was responsible for the virus protection, the IgG or IgM were removed by passing the BALF through a protein G- and anti-IgM antibody-Sepharose column, respectively (Supplementary Fig. 2). The absorption of IgG, but not IgM abolished the protective activity against A/PR/8/1934 (Fig. 2a), indicating that the IgG

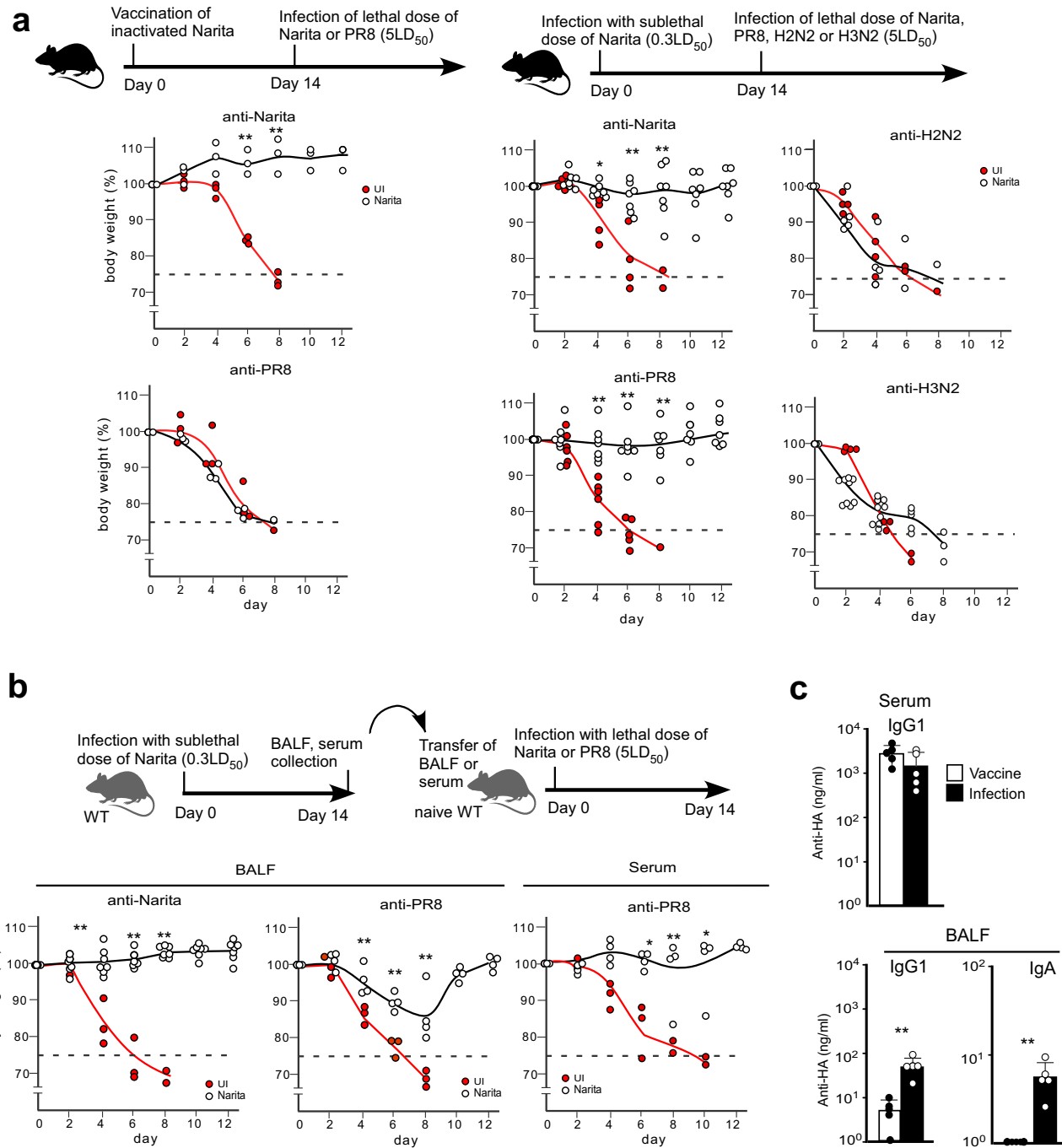

**Fig. 1 Live virus infection confers broadly protective humoral responses. a** Top panel indicates the design of the experiments for protection against heterotypic virus infection. Mice immunized by intraperitoneal vaccination with an inactivated whole virus (left $n = 3$) or by intranasal infection (right) of A/Narita/1/2009 were intranasally challenged with a lethal dose of A/Narita/1/2009 ($n = 5$), A/PR/8/1934 ($n = 7$), A/Okuda/1957 (H2N2 $n = 5$), or X31 (H3N2 $n = 9$) at 14 days post-immunization. The body weight changes were analyzed in the mice with or without (UI marked with red dots; $n = 3$–6) immunization. **b** Top panel indicates the design of the passive transfer experiment. Mice were intranasally administered BALF or serum (60 μg/ml anti-Narita IgG) from mice infected with A/Narita/1/2009 followed by intranasal challenge with a lethal dose of A/Narita/1/2009 (left $n = 6$) or A/PR/8/1934 (middle and right $n = 4$ for each). The body weight changes were analyzed in the mice administered BALFs from uninfected (UI marked with red dots; $n = 3$-4) or infected mice. The mice whose body weight fell below 75% of the starting weight were euthanized. **c** A/H1N1pdm09 HA-specific serum IgG1 (top) and BALF IgG1 and IgA titers were measured in mice intranasally immunized by vaccination using inactivated whole A/Narita/1/2009 (white bars; $n = 5$) or by sublethal infection with A/Narita/1/2009 (black bars; $n = 5$). UI represents unimmunized. Bars represent mean; *$p < 0.05$, **$p < 0.01$ by unpaired, two-tailed $t$-test (UI group with the other group in **a** and **b**, Vaccine group with Infection group in **c**). All error bars represent SEM.

class of antibodies was the primary component of the broadly protective Abs in the respiratory tract.

We further explored which respiratory tract region is essential for the induction of broadly protective Abs after nasal infection.

To distinguish between infection of the upper respiratory tract and the deep lung, we attempted to control different arrival sites of the virus by changing the amount of liquid while keeping the virus titer constant; 5 μl was used for the upper airway and 50 μl

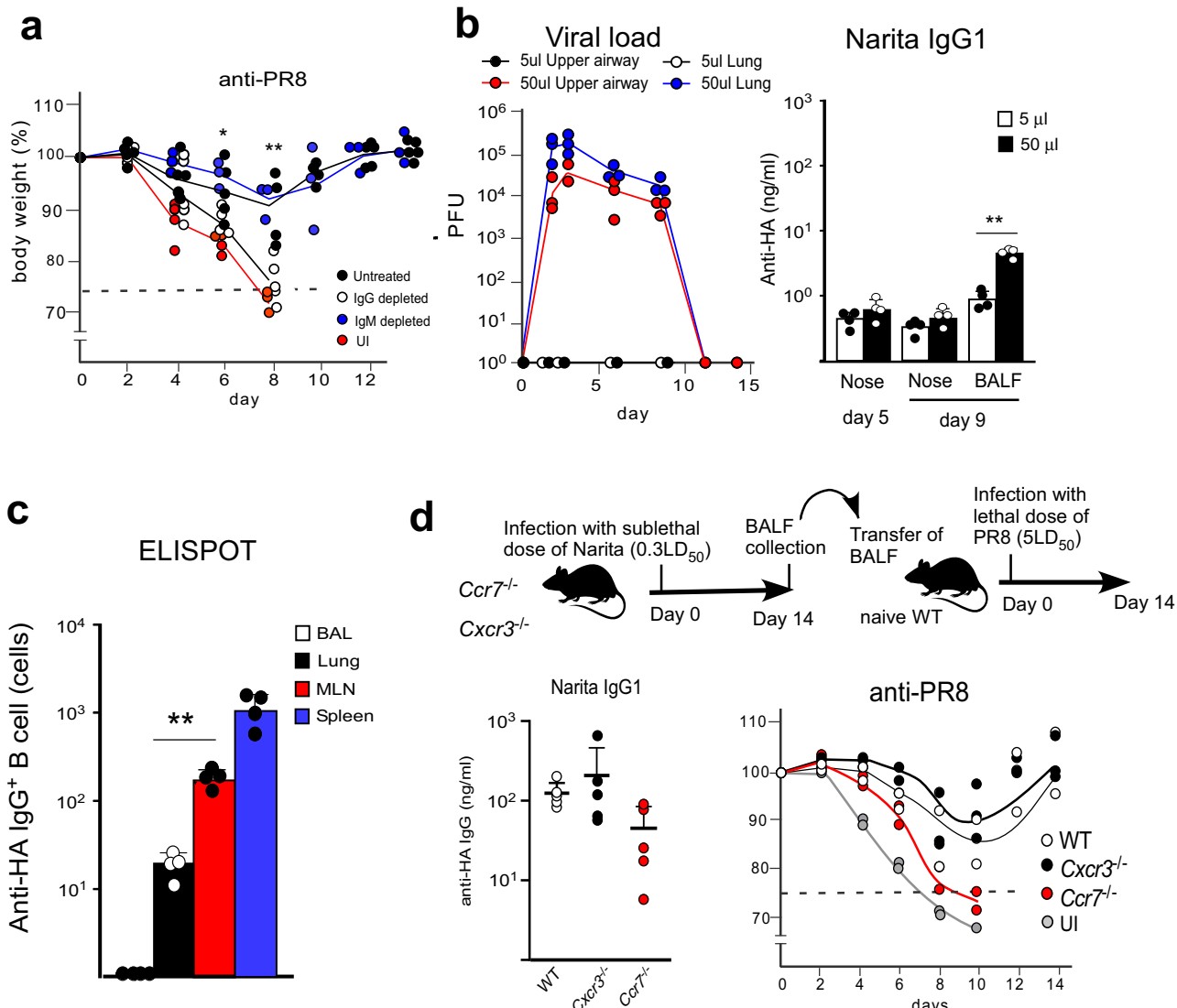

**Fig. 2 Anti-HA IgG responses in MLN are associated with broad influenza virus protection. a** Mice intranasally transferred with untreated (black dots; $n = 5$), IgG-depleted (white dots; $n = 6$), or IgM-depleted (blue dots; $n = 3$) BALF from A/Narita/1/2009 infected mice were challenged intranasally with a lethal dose of A/PR/8/1934. The data indicate body weight change of uninfected mice (red dots; UI $n = 3$) and the transferred mice with BALF. Missing symbols mean that the mice were euthanized, as described in Fig. 1b. **b** Kinetics of viral load (PFU) in the upper airway (black and red dots) and deeper lung (white and blue dots) were examined in mice infected with 5 or 50 μl of A/Narita/1/2009 suspension (left $n = 3$ for each). The titer of A/H1N1pdm09 HA-specific IgG1 in the nasal wash and BALF at day 5 and day 9 were determined by ELISA (right, white bars and black bars for 5 and 50 μl, respectively; $n = 4$ for each). **c** Numbers of A/Narita/1/2009-HA-specific IgG producing B cells in BALF (white), lung (black), MLN (red), or spleen (blue) from infected mice at 14 day were determined by ELISPOT ($n = 4$ for each). **d** IgG1 antibody titers specific for or A/Narita/1/2009-HA in BALF from WT (white), $Ccr7^{-/-}$ (red), or $Cxcr3^{-/-}$ (black) ($n = 5$ for each) mice were measured at day 14 post-infection with A/Narita/1/2009 (left). Top panel indicates the design of the BALF transfer experiment for protective antibodies in $Ccr7$- or $Cxcr3$-deficient mice. Mice intranasally administered with BALF from WT, $Ccr7^{-/-}$ or $Cxcr3^{-/-}$ mice infected with A/Narita/1/2009 (WT $n = 2$, $Ccr7^{-/-}$ $n = 2$, $Cxcr3^{-/-}$ $n = 3$) were challenged intranasally with a lethal dose of A/PR/8/1934. The body weight change of mice was monitored until 14 days post-infection. Missing symbols mean that the mice were euthanized, as described in Fig. 1b. UI represents unimmunized. Bars represent mean; *$p < 0.05$, **$p < 0.01$ by unpaired, two-tailed $t$-test (Untreated group with IgG depleted group in **a**, 5 μl group with 50 μl group in **b**, Lung group with MLN group in **c**). All error bars represent SEM.

for the deep-lung infection. The 50 μl condition led to a high viral load in the upper and lower tract, indicating that this system effectively introduced the live virus into the deep-lung (Fig. 2b). Infection in the upper and lower tract resulted in higher anti-HA IgG levels in the BALF (Fig. 2b).

To identify which secondary lymphoid organ is responsible for the production of anti-HA IgG antibodies, we investigated the localization of the B cells producing anti-HA IgG. The MLN and spleen showed many IgG+ AFCs in response to influenza HA, indicating that the MLN was a major secondary lymphoid organ

responsible for the anti-HA IgG response (Fig. 2c)[25]. CCR7 and CXCR3 are critical chemokines for T cell migration into the LN, and CXCR3 is a $T_H1$ chemokine receptor that acts through binding the CXC chemokines CXCL9, CXCL10, and CXCL11. The deletion of CCR7, but not CXCR3, resulted in reduced IgG antibody responses and protection against the A/PR/8/1934 challenge. Therefore, T cell migration into the MLN is crucial for the anti-virus antibody responses (Fig. 2d). These results suggested that IgG is the main isotype of the protective antibodies in the lung response to the influenza virus and that these IgG

responses are initially induced in the MLN associated with the lower respiratory tract and then become systemic.

**Natural infection induces IgG Abs recognizing sheared epitopes.** There seems to be a clear difference in antibody breadth between the inactivated vaccine and natural virus infection. We examined the binding to the HA antigens on A/Narita/1/2009 or

A/PR/8/1934 of membrane-bound IgG on plasmablasts sorted from MLN. The binding analysis showed the presence of A/Narita/1/2009-HA single binding (single) cells and dual binding (dual) cells that recognize HA antigen of both A/Narita/1/2009 and A/PR/8/1934. The dual-binding cells were found in the mice infected with live A/Narita/1/2009 but were much fewer in the inactivated vaccine recipients (Fig. 3a). More interestingly, the

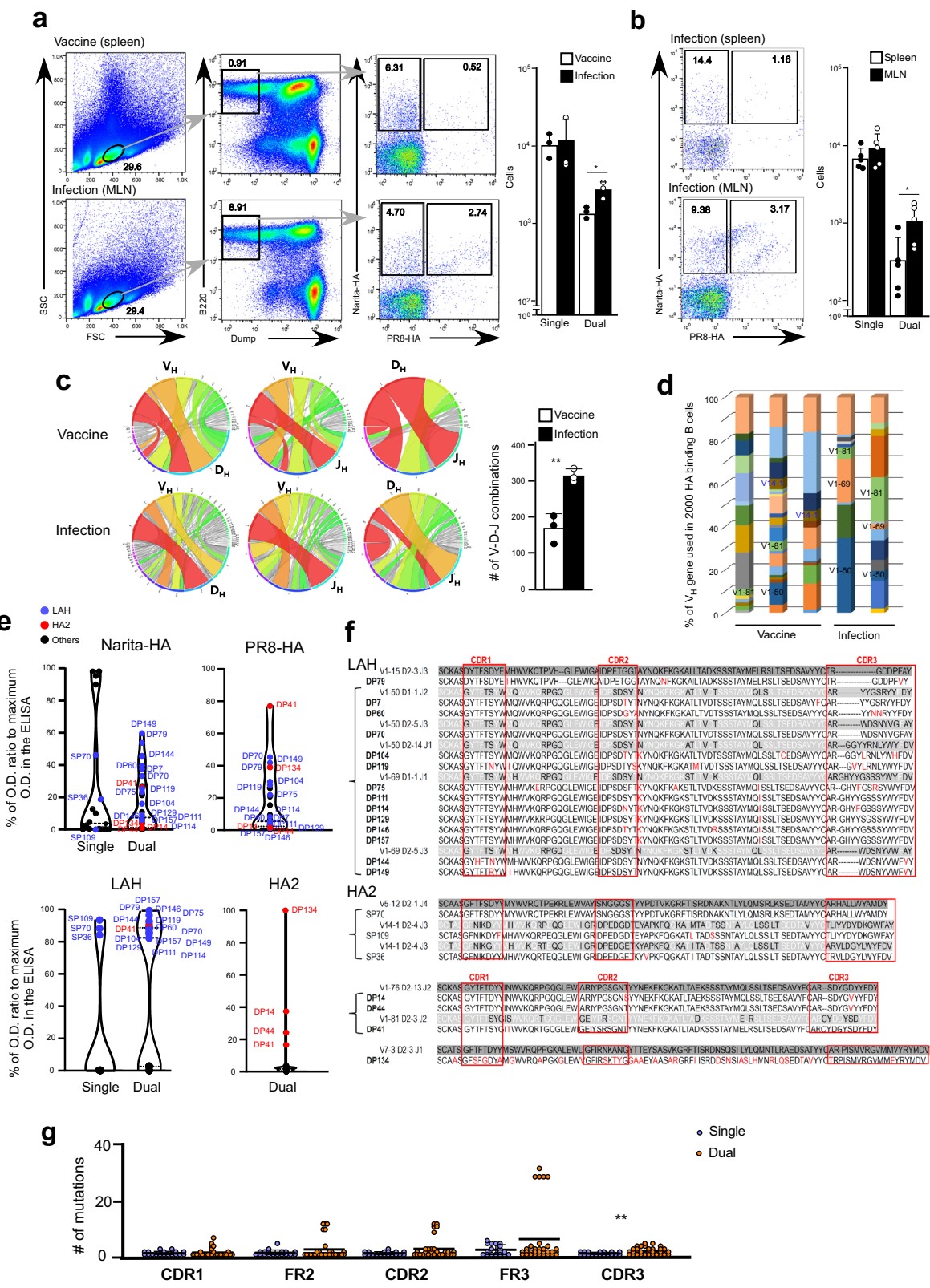

**Fig. 3 Selection of broadly reactive IgG is regulated differently from strain-specific IgG. a** B220[+]IgM[−]IgD[−]Dump[−] B cells were obtained from spleen of the mice immunized with inactivated A/Narita/1/2009 (Vaccine, white bars; $n = 3$) and from MLN in the A/Narita/1/2009 infected mice (Infection, black bars; $n = 3$). Binding to A/Narita/1/2009-HA and/or A/PR/8/1934-HA was determined by flow cytometry analysis. **b** The data indicate binding to HA in flow cytometry analysis (left) and the number of single or dual-binding B cells (right) in spleen (white bars) or MLN (black bars) from the mice infected with A/Narita/1/2009 ($n = 5$). **c** Narita-HA binding B cells (2000 cells each) were isolated from vaccinated mice (spleen, white bar; $n = 3$) and infected mice (MLN, black bar; $N = 3$). The V-D-J combinations are shown by connected color bands between V and D (left), V and J (middle), and D and J (right). **d** $V_H$ gene usage of A/Narita/1/2009-HA binding B cells in spleen from three vaccinated mice (Vaccine $n = 3$) or MLN (Infection $n = 2$) from two infected mice with A/Narita/1/2009. Total RNA isolated from sorted A/Narita/1/2009-HA binding B cells served as the template for cDNA synthesis followed by PCR to amplify IgG variable region genes. Amplified IgG gene libraries were sequenced by MiSeq. The IgG sequences were analyzed with the ImMunoGeneTics (IMGT) HighV-QUEST. **e** Seven hundred twenty HA probe[+] B cells were single sorted and cultured on NB21.2D9 feeder cells with IL-4 in individual wells. After 20 days of culture, 303 IgG-producing cells were obtained. The binding of antibodies produced by these cells to A/Narita/1/2009-HA, A/PR/8/1934-HA, A/H1N1pdm09 derived LAH (blue), or rHA2 (red) was measured by ELISA. Finally, reliable binding and $V_H$ genetic information were obtained for 101 clones. Black circles indicate clones showing no binding to LAH or rHA2. Y-axis represents the ratio of the $OD_{450}$ value in the sample vs the $OD_{450}$ value obtained with IgG [anti-Narita-HA (1 μg/ml), anti-PR8-HA (1 μg/ml), anti-LAH (1 μg/ml), and rHA-2 (3 μg/ml)]. **f** Amino acid sequences encoded by $V_H$ genes in single or dual-HA binding B cell clones in the region from CDR1 to CDR3 are shown with germline amino acid sequence (unhighlighted). Red boxes indicated CDR regions. The positions where amino acid substitutions occur are indicated by the red font. **g** Number of amino-acid mutations in the region extending from CDR1 to CDR3 of single or dual-binding B cell clones (single, blue dots; $n = 18$, dual, red dots; $n = 31$) as determined by referring to the germline sequences of the corresponding $V_H$ in the IMGT database. *$p < 0.05$, **$p < 0.01$ by unpaired, two-tailed $t$-test (Vaccine group with Infection group in **a** and **c**, Spleen group with MLN group in **b**, Single group with Dual group in **g**). All error bars represent SEM.

---

dual-binding cells emerged only in the MLN but not in the spleen of infected mice, suggesting that the broadly protective Abs were the result of a local rather than a systemic immune response (Fig. 3b). In contrast, vaccination by intraperitoneal and intramuscular administration was not sufficient to activate the immune response in the MLN. We further analyzed the BCR repertoire of the antigen-binding B cells and compared it between the inactivated vaccine (splenic B cells) and nasal infection (MLN B cells). BCR sequencing analysis revealed that the HA-binding IgG1[+] B cells in the infected mice had a different $V_H$ repertoire and a larger number of V-D-J combinations than the splenic B cells from the mice vaccinated with the inactivated virus (Fig. 3c, d). These results suggest that the dual-specific antibodies induced by live virus infection expressed a different repertoire of BCRs.

To further assess differences in the BCR repertoire between single-specific and dual-specific IgG antibodies, individual B cells obtained from the MLN of day 14 post-infection were sorted and expanded in mouse single B cell cultures (Nojima cultures)[26]. Among a total of 720 clones examined, 303 were IgG-producing B cells, and 49 clones produced anti-HA IgG antibodies at levels of more than 50 μg/ml. Binding analysis to the native HA antigens of A/Narita/1/2009 and A/PR/8/1934 indicated that 19 clones secreted single-specific and 30 clones secreted dual-specific antibodies (Fig. 3e). Fifteen clones of dual-specific B cell bound to a synthetic peptide corresponding to the long alpha-helix (LAH) of A/Narita/1/2009 HA that is generally hidden in the native form, and 7 out of 15 clones used V1-50-D2-14-J1 or V1-69-D1-1-J-1 (Fig. 3e, f).

In general, the antibodies targeting a common epitope on the HA stalk domain are rare because these epitopes are also hidden in the native form. However, we found four clones binding to the stalk domain and reactive with recombinant trimeric HA2 (rHA2) of the H1 subtype (Supplementary Fig. 3). Clone DP14 and 44 used the same $V_H$, D, and $J_H$ segments, V1-76-D2-13-J2, and bound to the rHA2 protein but not to the H1-derived LAH (Fig. 3e, f). The highest binding clone, DP134 used V7-3-D2-3-J1 and interestingly had undergone extensive SHM (Fig. 3f). Previous work indicated that SHM is likely not essential for generating the antibodies targeting the HA stalk domain[27]. Indeed, the number of overall mutations in $V_H$ genes of the dual-specific Abs was comparable to the single-specific Abs (Fig. 3g). Moreover, $V_H$ sequences in five of the dual-specific antibody clones retained germline sequences in their CDR regions (Fig. 3f). Therefore, these results suggested that nasal infection with A/

Narita/1/2009 induced broadly protective Abs that recognize the common epitopes on the HA stalk domain and LAH. Interestingly, pre-existing BCR repertoires are responsible for the recognition of heterotypic common epitopes on HA.

**Virus replication is required for broadly protective antibody**. The above data raise another question of how the nasal infection with live virus generates the broadly protective Abs. We speculated that virus replication would be a key process to explain this outcome. TMPRSS2 is a critical serine protease that proteolytically activates the HA protein of the H1N1 virus in mice[28]. Indeed, mice lacking TMPRSS2 have a defect in virus replication in the respiratory tract and are highly resistant to A/Narita/1/2009 infection (Fig. 4a). We decided to take advantage of the viral replication defect in *Tmprss2*-deficient mice to test whether virus replication is required to induce broadly protective IgG antibodies. Nasal administration with a low dose (sublethal dose 0.3LD$_{50}$) of live A/Narita/1/2009 failed to generate IgG antibodies specific for A/Narita/1/2009 HA (Fig. 4b). However, this defect could be explained by the insufficiency of the antigen dose required for T and B cell activation. Thus, the virus dose was increased up to 10,000-fold higher than the sublethal dose (3000LD$_{50}$). Nasal administration of *Tmprss2*-deficient mice with the high dose of live A/Narita/1/2009 virus promoted detectable levels of IgG antibodies, which were 10-fold lower than in the sufficient mice infected with a sublethal dose (Fig. 4b). We analyzed the protective ability of the sera by a passive transfer after normalization of serum amounts based on the anti-Narita-HA titer. The sera from the high dose TMPRSS2-deficient mice protected from A/Narita/1/2009 infection but not from A/PR/8/1934 infection (Fig. 4c). Indeed, *Tmprss2*-deficient mice showed a marked reduction of dual-specific antibodies (Fig. 4d). Moreover, a similar protection pattern was observed in mice treated with the virus replication blocker oseltamivir during the live Narita infection (Supplementary Fig. 4). These results indicated that virus replication occurring during nasal infection is a crucial process for generating the broadly protective Abs, suggesting that the virus replication process is tightly associated with a structural change in the virus HA that exposes rare antigenic epitopes.

**The broadly reactive Abs depend on GC responses**. The B cell memory response is thought to be critical for introducing SHM in the in Abs, and the acute infection seems to induce a B cell

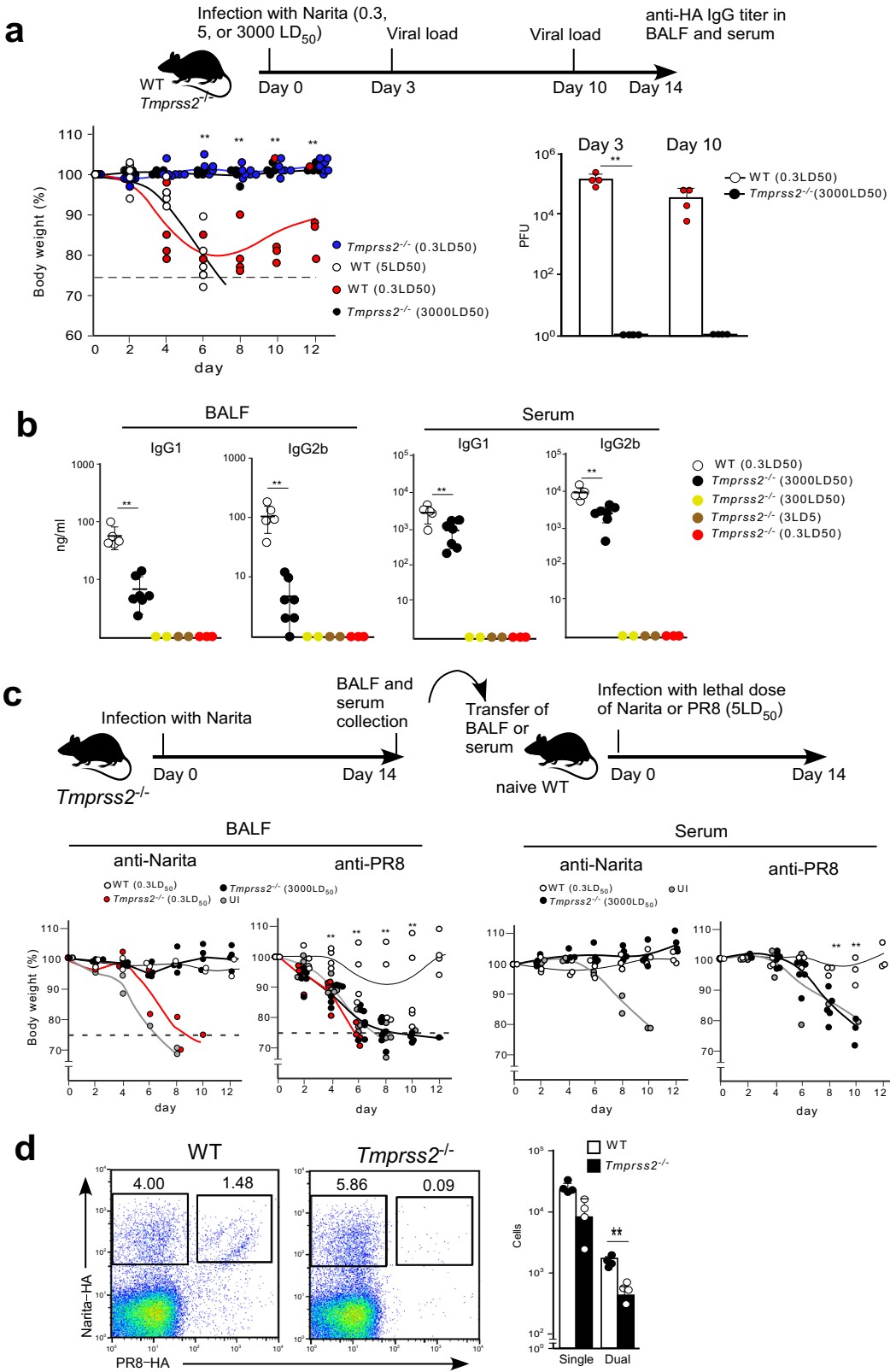

memory response. Indeed, we observed CD38+ memory B cells in the MLN at 10-days post-infection (Supplementary Fig. 5) and the infection conferred a broadly protective response for 40 days (Supplementary Fig. 6) . Thus, we asked whether the production of broadly protective Abs against the common HA epitopes depends on GC-dependent memory responses. Mice lacking $T_{FH}$ cells and GCs were prepared by deletion of *lox*P-flanked *Bcl6*

alleles (*Bcl6*^f/f) by Cre recombinase expressed from the CD4+ T-cell-specific *Cd4* promoter (*Bcl6*^f/f*Cd4-Cre*; referred to as *Bcl6*^ΔT) or the B-cell-specific promoter of the gene encoding the BCR signaling subunit CD79A (*Bcl6*^f/f*Cd79a-Cre*; referred to as *Bcl6*^ΔB). We analyzed the protective activity of BALF or sera against A/Narita/1/2009 or A/PR/8/1934 virus by the passive transfer system. In our previous work, the A/Narita/1/2009

**Fig. 4 Virus replication in the lower respiratory tract induces broadly protective IgG responses. a** Top panel indicates the design of the A/Narita/1/2009 infection experiment in $Tmprss2^{-/-}$ mice. Control C57BL/6 (WT) and $Tmprss2^{-/-}$ mice were infected with A/Narita/1/2009 (WT: 5LD$_{50}$ $n = 4$ (white), 0.3LD$_{50}$ $n = 4$ (red), $Tmprss2^{-/-}$: 0.3LD$_{50}$ $n = 7$ (blue), 3000LD$_{50}$ $n = 4$ (black)) (left). Missing symbols mean that the mice were euthanized, as described in Fig. 1b. The viral load (PFU) in the lungs at 3- or 10-day ($n = 4$ for each) post-infection was determined by MDCK culture (right). **b** IgG1 and IgG2b titers against Narita-HA in BALF and serum from WT (white, $n = 5$) and $Tmprss2^{-/-}$ mice at 14 days post-infection with A/Narita/1/2009 (3000LD$_{50}$ $n = 7$ (black), 300LD$_{50}$ $n = 2$ (yellow), 3LD$_{50}$ $n = 2$ (brown), 0.3LD$_{50}$ $n = 3$ (red)). **c** Top panel indicates the design of the passive transfer experiment in $Tmprss2^{-/-}$ mice. Mice intranasally transferred with BALF or serum (60 μg/ml anti-Narita IgG) from WT (white) (anti-Narita $n = 3$, anti-PR8 $n = 4$) or $Tmprss2^{-/-}$ (3000LD$_{50}$ anti-Narita in BALF $n = 3$, anti-PR8 in BALF $n = 8$, anti-Narita in serum $n = 5$, anti-PR8 in serum $n = 6$ (black), 0.3LD$_{50}$ anti-Narita in BALF $n = 2$, anti-PR8 in BALF $n = 4$ (red)) mice infected with A/Narita/1/2009 were challenged intranasally with a lethal dose of A/Narita/1/2009 (left) or A/PR/8/1934 (right). Missing symbols mean that the mice were euthanized, as described in Fig. 1b. **d** Control WT (white bars, $n = 4$) and $Tmprss2^{-/-}$ (black bars, $n = 4$) mice were infected with A/Narita/1/2009, and binding to Narita-HA or PR8-HA was analyzed at 14 days post-infection. UI represents unimmunized. Bars represent mean; *$p < 0.05$, **$p < 0.01$ by unpaired, two-tailed $t$-test. WT (0.3LD$_{50}$) group was compared with $Tmprss2^{-/-}$ (3000LD$_{50}$) group. All error bars represent SEM.

protection induced by the inactive vaccine was unaffected by $Bcl6$ deficiency[1]. In contrast, the protection against A/PR/8/1934 was abolished in recipients of BALF or sera from the $Bcl6^{\Delta T}$ and $Bcl6^{\Delta B}$ mice (Fig. 5a), which showed a marked reduction of GC formation in the MLN (Supplementary Fig. 7a). The GC-deficient mice also showed a marked decrease in the IgG antibody titer against A/PR/8/1934-HA and in the number of dual-specific B cells (Fig. 5b, c). Nevertheless, there was no significant reduction in the response to nuclear protein (Supplementary Fig. 7b). These results indicated that the dual-reactive anti-HA antibody production in the MLN during acute infection is mostly dependent on T$_{FH}$ cell-regulated GC responses (Supplementary Fig. 7c). However, SHM and affinity maturation are not necessary for the broadly protective Abs, because germline BCRs reacted with the hidden epitopes on the HA stalk and LAH (Fig. 3e), suggesting that the central role of the GC response would be the expansion of the small number of B cells recognizing the common epitopes.

**IL-4 controls GC-B cell expansion in virus responses.** The generation of the broadly protective Abs against the common HA epitopes in the MLN requires the virus replication process in the lower respiratory tract. Thus, viral replication may be necessary to create a unique cytokine environment that efficiently promotes broadly protective Abs. To examine the involvement of pro-inflammatory cytokines and chemokines, we examined IgG antibody titers in BALF against A/PR/8/1934-HA after the A/Narita/1/2009 virus infection. IL-6 deficiency reduced both the single- and dual-reactive IgG antibodies and the protective response against the A/PR/8/1934, indicating that IL-6 was not the factor that specifically controlled the broadly protective Abs (Fig. 6a, b).

Thus, we next focused on T$_{FH}$ cell-derived cytokines, IFN-γ, IL-4, and IL-21, which are produced by T$_{FH}$ cells in the MLN of infected mice (Supplementary Fig. 8). $Ifng^{-/-}$, $Il4^{-/-}$ or $Il21^{f/f}Cd4$-$Cre$ (referred to as $Il21^{\Delta T}$) mice were infected with A/Narita/1/2009 virus, and then we tested whether their BALFs were protective against challenge with A/PR/8/1934. The $Il4^{-/-}$ mice, but not in the $Ifng^{-/-}$ and $Il21^{\Delta T}$ mice, showed a marked attenuation of the protection against A/PR/8/1934 infection (Fig. 6c). More importantly, the $Il4^{-/-}$ mice also showed a severe reduction in generation of GC B cells after A/Narita/1/2009 infection, and $Il21^{\Delta T}$ mice showed a partial reduction of GL-7$^{hi}$ GC-B cells, while the $Ifng^{-/-}$ mice had normal GC development (Fig. 6d). To confirm the importance of IL-4 in B cell responses, we used mice with a B cell-specific deficiency in the IL-4 receptor α chain gene, $Il4ra^{f/f}Cd79a$-$Cre$ (referred to as $Il4ra^{\Delta B}$). Mice transferred with antibodies from $Il4ra^{\Delta B}$ mice were susceptible to the A/PR/8/1934 challenge (Fig. 6c). Consistently, the $Il4^{-/-}$ and $Il4ra^{\Delta B}$ mice infected with A/Narita/1/2009 showed a significant

reduction in dual-specific B cells and antibodies in the BALF (Fig. 6e).

To directly prove the role of the IL-4 signal in the A/Narita/1/2009 infection, we performed bone marrow chimera transfer (BMT) studies. CD45.2 bone marrow cells from $Il4ra^{\Delta B}$ mice were transferred into irradiated CD45.1 hosts, followed by A/Narita/1/2009 infection. IL-4R-deficient B cells showed less proliferation and a marked reduction of dual-specific B cells in the GC (Fig. 7a). These results indicate that IL-4 signaling is essential for the GC formation and the production of the broadly protective Abs.

A final important question is how IL-4 contributes to the GC response. T$_{FH}$-B cell interactions in the GC light zone (LZ) trigger signals that control B cell activation and these signals are transmitted through the BCR and CD40, which activate the NFκB and PI3K-Akt-mTOR pathways[29]. The T$_{FH}$-mediated signals positively select antigen-specific B cells, then these LZ GC-B cells can transit into the dark zone (DZ) to expand. In this context, activation of the PI3K-Akt-mTOR pathway promotes the LZ GC-B cells by transiently downregulating the expression of $Foxo1$, a critical transcriptional factor for DZ formation[30]. Therefore, we speculated that IL-4 signaling regulates activation of the PI3K-Akt-mTOR pathway. To test this possibility, we evaluated phosphorylation of ribosomal protein S6, a target of mTORC1, on serine 235 and 236 using the OT-II T cell transfer system. Mice transferred with IL-4-deficient OT-II T cells were significantly impaired in the emergence of pS6$^+$B220$^+$ cells (Fig. 7b and Supplementary Fig. 9). The absence of IL-4 production by T$_{FH}$ cells also dampened the expression of c-$Myc$, c-$Myb$, $Bcl2$, $Bcl6$, and AID (Fig. 7c) and the expression of Ki67, a marker of cell proliferation, in the GL-7$^+$B220$^+$ B cells (Fig. 7d). These results indicated that the IL-4 signaling controls the expansion of the rare GC-B cells recognizing the shared epitopes via the mTOR pathway and B cell proliferation.

**Discussion**

The present work demonstrated that infection and vaccination are quite different in terms of the breadth of antibodies generated. Unlike inactivated vaccines, nasal infection with influenza virus is a promising strategy to induce broadly protective Abs, targeting overlapping epitopes shared by different viral serotypes. We here propose that two essential processes during infection are required for the generation of the broadly protective Abs. Virus entry through the respiratory tract and its replication is necessary for the production of the broadly protective Abs. In this case, virus replication causes a structural change in the viral HA that exposes to B cells the shared epitopes that are hidden in the native HA structure. BCR sequencing analysis indicated that pre-existing B cell clones might be responsible for the recognition of the sheared epitopes. These B cell populations must be amplified by the GC

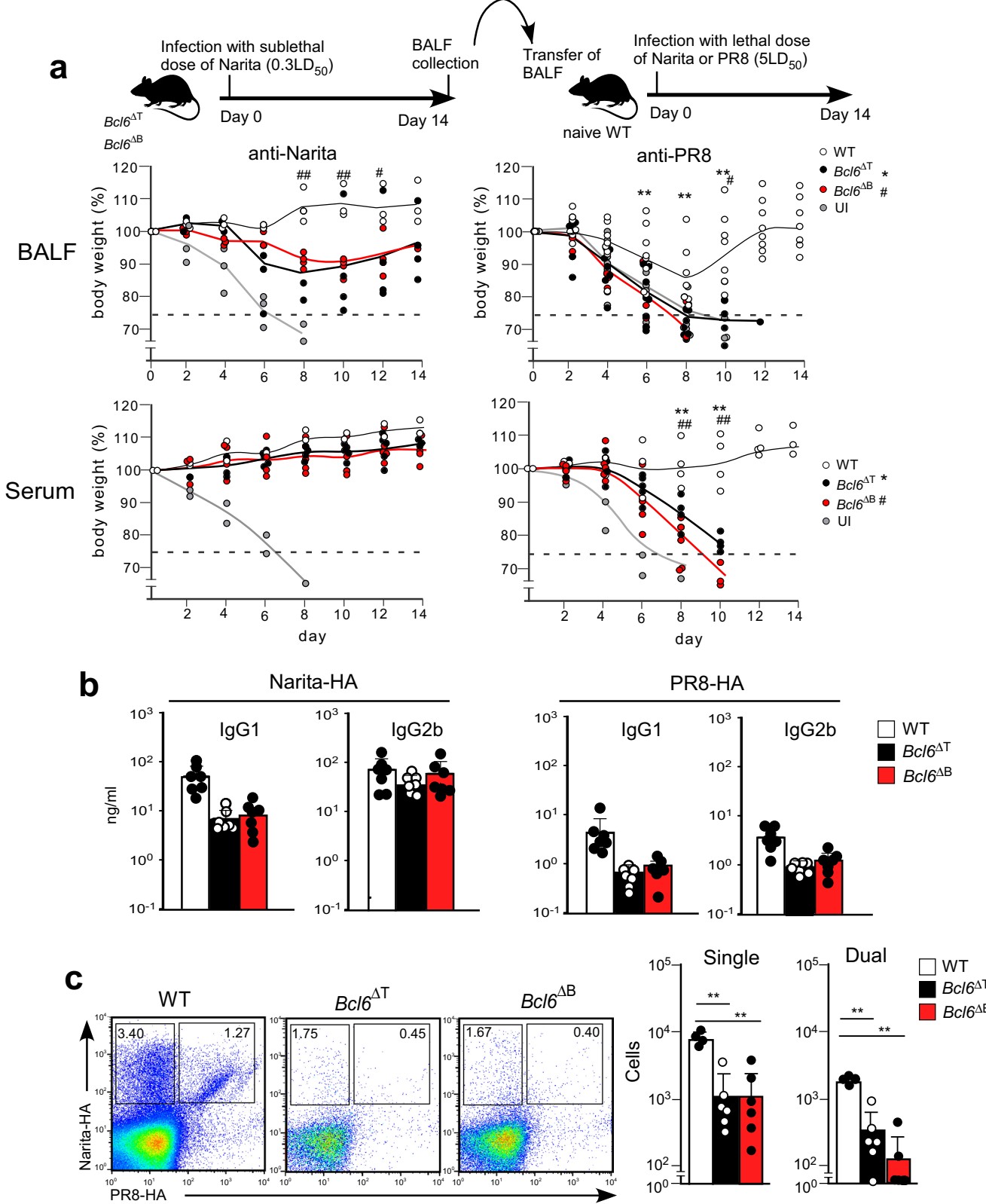

response to efficiently generate the broadly protective Abs in the draining MLN. In this context, IL-4 had an intrinsic role to efficiently expand GC-B cells via catabolic and anabolic metabolism. Therefore, $T_{FH}$ cell-derived IL-4 is essential to facilitate B cell proliferation in the GC that efficiently expands the clone size of the minor population of B cells recognizing heterotypic HA epitopes.

We have previously demonstrated that vaccination of C57BL/6 J mice with inactivated A/Narita/1/2009 virus and a highly pathogenic avian virus, H5N1, induced GC-independent anti-HA IgG2 antibodies that were mainly controlled by $T_H1$ cells[1]. However, the present data demonstrated that these types of vaccines induced strain-specific antibody, that were effective only for the virus strain used for vaccination, consistent with previous

**Fig. 5 GC function is required for cross-strain protective humoral responses. a** Top panel indicates the design of the BALF transfer experiment for protective antibody in Bcl6-deficient mice. Mice intranasally administered with BALF or serum (60 μg/ml anti-Narita IgG) from WT (white), $Bcl6^{\Delta T}$ (black) or $Bcl6^{\Delta B}$ (red) mice infected with A/Narita/1/2009 were challenged intranasally with a lethal dose of A/Narita/1/2009 (left) (BALF WT $n = 3$, BALF $Bcl6^{\Delta T}$ $n = 4$, $Bcl6^{\Delta B}$ $n = 4$, Serum WT $n = 3$, $Bcl6^{\Delta T}$ $n = 5$, $Bcl6^{\Delta B}$ $n = 6$) or A/PR/8/1934 (right) (WT $n = 8$, $Bcl6^{\Delta T}$ $n = 10$, $Bcl6^{\Delta B}$ $n = 4$, Serum WT $n = 4$, $Bcl6^{\Delta T}$ $n = 5$, $Bcl6^{\Delta B}$ $n = 6$). The data indicate body weight change. Missing symbols mean that the mice were euthanized, as described in Fig. 1b. **b** A/ H1N1pdm09-HA- and A/PR/8/1934HA-specific IgG1 and IgG2 titers in BALF from WT (white bars), $Bcl6^{\Delta T}$ (black bars), and $Bcl6^{\Delta B}$ (red bars) mice ($n = 7$ each) at 14 days post-infection with A/Narita/1/2009 were measured by ELISA. **c** The data indicate single or dual-binding B cells in the $B220^+IgM^-IgD^-Dump^-$ population from MLN of WT (white bars, $n = 4$), $Bcl6^{\Delta T}$ (black bars, $n = 5$), and $Bcl6^{\Delta B}$ (red bars, $n = 7$) mice. UI represents unimmunized. Bars represent mean; *,#$p < 0.05$, **,##$p < 0.01$ by unpaired, two-tailed $t$-test (WT group with other groups respectively). All error bars represent SEM.

reports that vaccination with the inactivated virus was less effective against a newly circulating strain[4]. In contrast, several lines of evidence indicate that antibody responses to natural infection are relatively broad and exhibit different immunodominance from those induced by the inactivated vaccine[2,7]. In humans, infection with A/Narita/1/2009 virus resulted in the broadly protective Abs recognizing other group-1 HA proteins[6,31]. Our mouse system similarly indicated that nasal administration of live A/Narita/1/2009 virus promoted the production of broadly protective Abs capable of recognizing a common HA epitope on heterotypic H1N1 viruses. These broadly protective Abs were predominantly generated in the draining MLN and distributed in serum and the lung. We used intramuscular vaccination in the thigh muscles but popliteal LN showed no dual-specific B cells (data not shown). Furthermore, the broadly reactive IgG Abs, but not strain-specific IgG Abs, were completely abolished in the mice lacking virus replication, suggesting that virus replication in the lower respiratory tract is required to expose the shared epitopes necessary for the generation of the broadly protective Abs.

However, it remains unclear why virus replication expands the breadth of the antibody response. A likely explanation is that conformational plasticity of the HA structure occurs during the life-cycle of the virus[32]. The conformational changes could allow exposure of the overlapping epitopes that are generally hidden in the native form of HA. Therefore, the overlapping epitopes may be presented to B cells in the MLN GCs. Our observation supports this hypothesis that lack of virus replication caused a marked reduction of dual-specific antibodies. It is also supported by the presence of overlapping epitopes for induction of the bnAbs[33] and the observation that dissociation of HA structure by acid treatment enhanced the generation of broadly reactive GC-B cells in secondary lymphoid organs[34].

Nasal infection with A/Narita/1/2009 predominantly induced antibodies that recognize the LAH domain, which contains conserved epitopes among HA subtypes, but this region is generally masked by the trimeric HA structure in its native state. Another class of dual-specific antibodies recognized the HA2 stalk region, which is less susceptible to antigenic changes[33]. Immunization with native HA trimers barely elicits LAH antibodies in humans or mice[34], and the stalk-specific antibodies are also rare and emerge late in the response because of the restricted accessibility of the membrane-proximal HA stalk domain[11]. However, in the nasal infection mouse model with A/ H1N1pdm09, the antibodies recognizing either the LAH or stalk domain were induced within the first two weeks and conferred broad protection against heterotypic influenza virus.

Sequencing results indicated that two major Ig $V_H$ genes, VH1-50 and VH1-69, combined with a relatively restricted Vκ gene repertoire, recognized the LAH peptide of H1 that had 50% homology with H3. More interestingly, five different clones of VH1-69-D1-J1-1 contained a quite similar and extended CDR3 region. Previous human work has indicated that rapidly emerging antibodies specific for the stalk region frequently use VH1-69

genes and exhibit heterosubtypic breadth within group 1 viruses[19]. The human VH1-69 genes retain the germline sequence and provide pre-existing immunity to the bnAb epitope on the HA stalk without SHM[20]. Our sequencing data support this idea that the broadly-reactive mAbs can arise from germline sequences, because the dual-specific antibodies against LAH and stalk domain had few mutations in their CDR regions. Therefore, we expect that GC responses against heterotypic HA antigen may have different roles other than SHM and affinity maturation of antibodies.

We found that IL-4 derived from $T_{FH}$ cells was necessary for efficient GC formation and for widening the breadth of influenza-specific antibody responses. IL-4 promotes class-switch recombination of IgG1 and IgE genes in mice and develops antibody-secreting cells via upregulation of the transcription factor Blimp-1[35,36]. Gaya's group has reported that lack of an early wave of IL-4 derived from NKT cells in the B cell follicle caused a significant reduction in the proportion of GC-B cells during influenza infection[37]. However, our data indicate the importance of a later wave of IL-4, which is mainly secreted by $T_{FH}$ cells. This is consistent with the report that GC migrated $T_{FH}$ cells are the most abundant source of IL-4 in secondary lymphoid organs[38]. Generation of the broadly protective Abs in the draining MLN was mostly dependent on $T_{FH}$-regulated GC responses after intranasal infection with A/Narita/1/2009 virus. These results suggest that IL-4 derived from $T_{FH}$ cells plays an intrinsic role in the GC-B cell response responsible for the broadly protective Abs.

B cell proliferation through induction of various target genes of the mTOR and expression of series of transcriptional factors critical for GC initiation and function, including c-Myc and AID was severely dampened in B cells lacking IL-4 signaling. These results explain previous studies showing that rapamycin treatment or a defect in the mTOR pathway in B cells resulted in a reduced GC response[39], and that IL-4 promotes the expression of BLIMP1, AID, and the co-stimulatory molecule CD40 by GC-B cells[38]. In this case, IL-4 is most effective for B cell expansion in the early wave of activation, prior to entering GC recycling, based on our finding that the number of $GL-7^+$ $CD38^{dull}$ $PD1^+$ GC-B cells was markedly decreased in $Il4ra^{\Delta B}$ mice. Therefore, we propose that $T_{FH}$-derived IL-4 has a rather preferential role in increasing the clone size of the pre-existing B cells that have a natural affinity for the bnAb epitope on the influenza virus HA antigen.

In conclusion, our data demonstrate that a distinct feature of the overlapping epitopes on the HA antigen between inactivated and live virus provides an opportunity to generate antibodies capable of binding to antigenically different HA viruses. Our data indicate that direct delivery of native virus antigen into airway epithelial cells in the lung could be a promising approach for the exposure of a sheared epitope that is typically hidden. This strategy may mimic the virus replication process that occurs in natural infection, thereby increasing the chances of expanding the clone size of very rare B cell clones able to recognize the common epitopes. In this context, IL-4 derived from $T_{FH}$ cells plays a

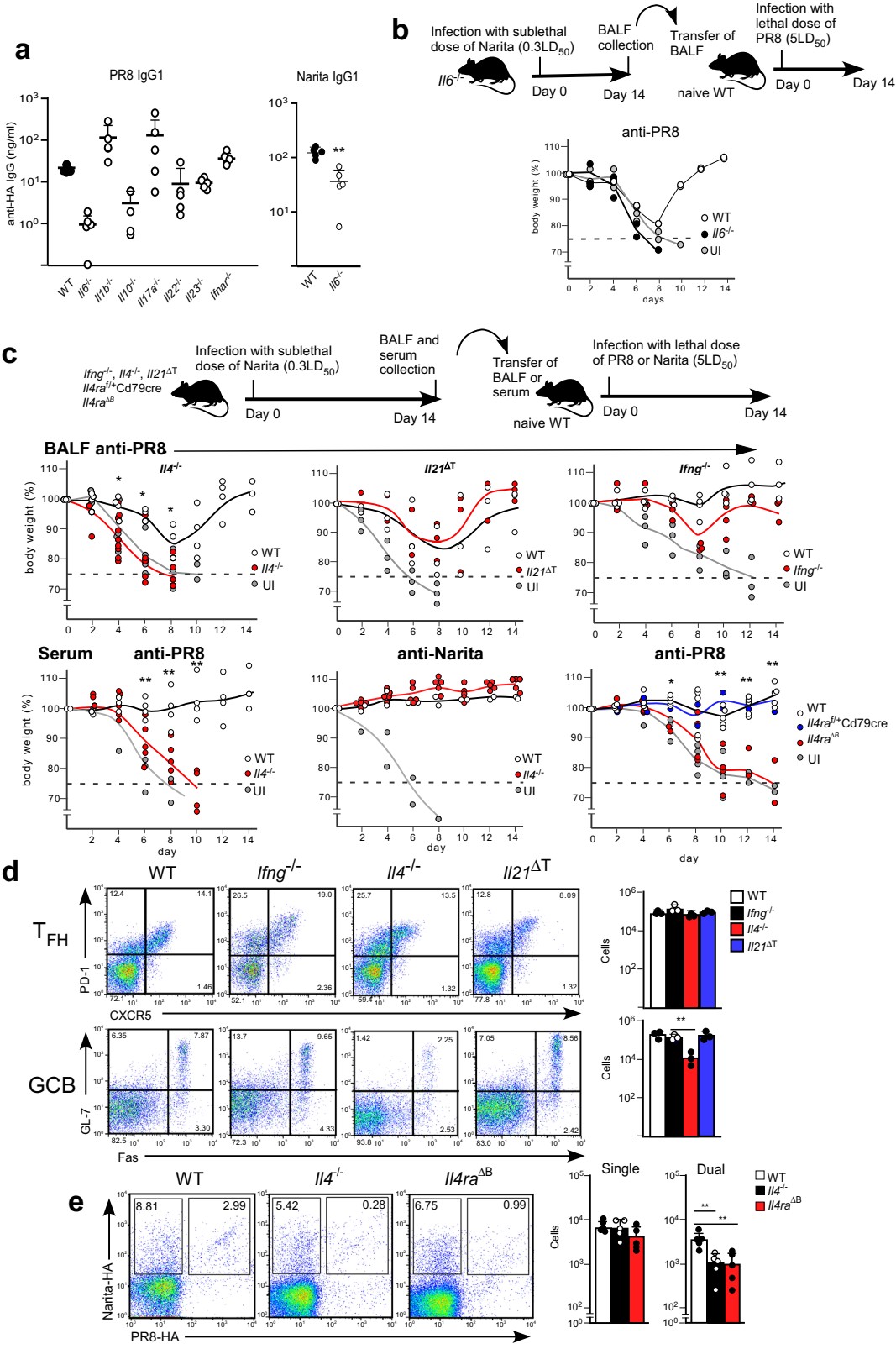

critical role in efficiently expanding such a B cell repertoire. Therefore, a combination strategy of the delivery of native virus antigen in the lung and the activation of IL-4 signaling might be a successful approach to generate the bnAb response.

## Methods

**Mice.** C57BL/6Jjcl mice were purchased from CLEA Japan (Meguro, Tokyo, Japan). Six to ten week old female mice were used for experiments. *Bcl6^f/f Cd4-Cre,* and *Cd79a-Cre* mice are described in a previous report[1]. IFN-γ Venus, *Il21* hCD2 and *Il4* hCD2 reporter mice, *Il21^f/f* mice and CNS2^−/− mice were generated as

**Fig. 6 IL-4 signaling is required for broadly protective IgG responses. a** IgG1 antibody titers specific for A/PR/8/1934-HA- or A/Narita/1/2009-HA in BALF from WT (black), $Il6^{-/-}$, $Il1b^{-/-}$, $Il10^{-/-}$, $Il17a^{-/-}$, $Il22^{-/-}$, $Il23^{-/-}$, $Ifnar^{-/-}$ (white, $n = 5$ for each) mice were measured at day 14 post-infection with A/Narita/1/2009. **b** Top panel indicates the design of the BALF transfer experiment for protective antibodies in $Il6^{-/-}$ mice. Mice intranasally administered with BALF from WT, $Il6^{-/-}$ mice infected with A/Narita/1/2009 (WT $n = 2$ (white), $Il6^{-/-}$ $n = 2$ (black)) were challenged intranasally with a lethal dose of A/PR/8/1934. The body weight change of mice was monitored until 14 days post-infection. **c** Top panel indicates the design of the BALF or serum transfer experiment. Mice intranasally transferred with BALF or serum (60 µg/ml anti-Narita IgG) from WT (white, $n = 2$–5), $Ifng^{-/-}$ (red, $n = 4$), $Il21^{\Delta T}$ (red, $n = 4$), $Il4^{-/-}$ (red) (anti-PR8 in BALF $n = 9$, in serum $n = 6$; anti-Narita in serum $n = 5$), $Il4ra^{f/+}Cd79cre$ (blue, $n = 2$), or $Il4ra^{\Delta B}$ (red, $n = 4$) mice infected with A/Narita/1/2009 were challenged intranasally with a lethal dose of A/Narita/1/2009 or A/PR/8/1934. Anti-A/Narita/1/2009 HA total IgG titers in serum were corrected to 60 µg/ml by dilution before the serum transfer. The body weight change of mice was monitored until 14 days post-infection. Missing symbols mean that the mice were euthanized, as described in Fig. 1b. **d** $T_{FH}$ cells (CD4$^+$ PD-1$^+$ CXCR5$^+$) and GC B cells (B220$^+$ GL-7$^+$ Fas$^+$) were analyzed in MLN from the A/Narita/1/2009 infected WT (white bars), $Ifng^{-/-}$ (black bars), $Il21^{\Delta T}$ (blue bars), or $Il4^{-/-}$ (red bars) ($n = 3$ for each) mice. **e** The data show the percentage and number of single and dual-binding B cells (B220$^+$IgM$^-$IgD$^-$Dump$^-$) in MLN from the A/Narita/1/2009 infected WT (white bars), $Il4^{-/-}$ (black bars), or $Il4ra^{\Delta B}$ (red bars) ($n = 5$ for each) mice. UI represents unimmunized. Bars represent mean; $*p < 0.05$, $**p < 0.01$ by unpaired, two-tailed $t$-test (WT group with other groups respectively). All error bars represent SEM.

---

previously described[1,40–42]. $Tmprss2^{-/-}$ mice were gift form Dr. M. Takeda (National institute of infectious diseases, Tokyo, Japan)[28]. $Il6^{-/-}$, $Il1b^{-/-}$, $Il10^{-/-}$, $Il17a^{-/-}$, $Il22^{-/-}$, $Il23^{-/-}$, $Ifnar^{-/-}$, and $Cd28^{-/-}$ are described elsewhere[43–50]. $Il23^{-/-}$ mice were obtained from Schering-Plough Biopharma. $Il4^{-/-}$ and $Il6^{-/-}$ mice were kindly provided by Dr. K. Manfred (ETH Zurich, Department of biology, Zurich, Switzerland). $Il4ra^{-/-}$ and $Il4ra^{f/f}$ mice were kindly provided by Dr. F. Brombacher (University of Cape Town, South Africa). $Ccr7^{-/-}$ and $Cxcr3^{-/-}$ mice were generated using the CRISPR/Cas9 system. Targeting constructs for $Ccr7$ or $Cxcr3$ containing guide RNA and Cas9 mRNA that were constructed with primers listed in Supplementary Table 1 were injected into fertilized mouse eggs. Genotyping of offspring was carried out by flow cytometry analysis using anti-CCR7 (clone: 4B12; 1:200 dilution, eBioscience, San Diego, CA, 12-1971-82) and CXCR3 (clone: CXCR3-173; 1:200 dilution, BioLegend, San Diego, CA, 126506) antibody. All gene targeted mice used in this study were on a C57BL/6 background. Control animals were bred separately. All mice used in this study were maintained under specific pathogen-free conditions and animal care was under the guidelines of the RIKEN Yokohama Institute.

**Viruses**. Influenza A virus (H1N1), A/PR/8/1934 was obtained from ATCC (Manassas, VA). A/Narita/1/2009 was propagated in MDCK cells (ATCC). Infection was carried out by intranasal injection of virus suspension under the anesthesia using isoflurane. A/Okuda/1957 (H2N2) and X31 (H3N2) adapted to mice were maintained by a passage in mice.

**Vaccination and protective assay**. For vaccination with inactivated virus, mice have intraperitoneally injected twice with inactivated virus particles (10 µg) conjugated with alum adjuvant at a 7 day interval. For live virus vaccination, virus suspension (sublethal dose, 0.3LD$_{50}$) in 50 µl of PBS was administrated nasally. To control the viral arrival point in Fig. 2b, we established a system to distinguish the infection site between the upper respiratory tract and the deep lung by changing the volume of PBS, while keeping the virus titer in the inoculum constant. For passive transfer, BALF was collected by bronchoalveolar washing and was concentrated up to 10 times with a centrifugal concentrator (Vivaspin 500, Sartorius, Göttingen, Germany). Serum HA IgG titers were normalized with 60 µg/ml of anti-A/Narita/1/2009 (Supplementary Fig. 10). Transfers of BALF or serum were carried out intranasally prior to virus challenge. For the in vivo protective assay, mice were given a lethal dose of influenza virus (5LD$_{50}$) intranasally at 14 days post-vaccination. The body weight changes were analyzed in the mice with or without vaccination. Mice whose body weight fell below 75% of the starting weight were euthanized.

**ELISA, ELISPOT and measurement of antibody affinity**. For the ELISA assay, the viral antigens of A/Narita/1/2009 or A/PR/8/1934 propagated in MDCK cells were captured on 96-well Maxisorp ELISA plates (Thermo Fisher, Waltham, MA) by anti-California (clone: RM01; 1:10,000 dilution, SinoBiological, Beijing, China, 86001-RM01) or PR8 HA antibody (clone: R107; 1:10,000 dilution, SinoBiological, 11684-R107), respectively. ELISA of LAH peptide and mini-HA, A/Narita/1/2009 HA 420-474 and recombinant HA2 (rHA2)[51] were used as LAH[52] and mini-HA antigens. IgG titers for the nuclear protein of influenza virus were measured by ELISA using His-conjugated recombinant Narita or PR8 nuclear protein (11675-V08B; SinoBiological) after capturing with an anti-His antibody. After blocking with 1% blockAce (KAC, Kyoto, Japan), antibody titers were assessed by anti-mouse IgG1 Abs (1:10,000 dilution, Southern Biotech, Birmingham, AL, 1070-01 and 1:10,000 dilution, Bethyl, Montgomery, TX, A90-105P), anti-mouse IgG2b Abs (1:10,000 dilution, Bethyl, A90-109P), anti-mouse IgG2c Abs (1:10,000 dilution, AVIVA System Bio, San Diego, CA, OASB01582), and anti-mouse IgA Abs (1:10,000 dilution, Bethyl, A90-103A and A90-103P). After washing, isotype-specific antibodies were visualized using HRP-conjugated antibody and OptiEIA substrate (BD Biosciences, San Jose, CA). For standard antibodies to assess Ig

subtype-specific antibody concentration, anti-H1N1 pandemic or seasonal HA IgG1, IgG2b, or IgA recombinant proteins were constructed based on two mAbs, NSP2[53] and S5V2-29[54]. For measurement of NP-specific antibody, antibody titer and affinity were assessed by the binding to either NP$_{19}$-BSA or NP$_1$-BSA. For ELISPOT assays, A/Narita/1/2009 antigens were captured by anti-Narita-HA (86001-RM01) on 96 well ELISPOT plates (MSHAN4B; Millipore). The purified B220$^+$ B cells from BALF, lung, MLN, or spleen of infected mice were cultured for 20 h, then the antibodies on the plate were detected with anti-mouse IgG-AP 1:5000 dilution, Proteintech, Rosemont, IL, SA00002-1).

**Histology**. Spleens were fixed with paraformaldehyde (4%) and frozen in OCT compound (Sakura Finetek Japan Co, Tokyo, Japan). Sections (5 µm) were stained with anti-B220 (clone: RA3-6B2; 1:100 dilution, Biolegend, 103208), CD4 (clone: RM4-5; 1:100 dilution, Biolegend, 100530), GL-7 (1:200 dilution, BD Biosciences, 553666), and pS6 (clone: N7-548; 1:200 dilution, BD Biosciences, 560434) mAbs after blocking (3% BSA) and permeabilization (0.1% Trironx-100). Images were acquired with a BZ-X700 (Keyence, Osaka, Japan).

**Flow cytometry**. Flow cytometric analysis was carried out with the following antibodies. anti-B220 (clone: RA3-6B2; 1:1000 dilution, Biolegend, 103227), CD3 (clone: 145-2C11; 1:200 dilution, Biolegend, 100304), CD4 (clone: GK1.5; 1:1000 dilution, Biolegend, 100423), CD5 (clone: 53-7.3; 1:200 dilution, Biolegend, 100604), CD8a (clone: 53-6.7; 1:200 dilution, Biolegend, 100704), CD11b (clone: M1/70; 1:200 dilution, Biolegend, 101204), CD11c (clone: N418; 1:200 dilution, Biolegend, 117304), CD19 (clone: 6D5; 1:1000 dilution, Biolegend, 115528), CD38 (clone: 90; 1:200 dilution, Biolegend, 102714), CD80 (clone: 16-10A1; 1:200 dilution, Biolegend, 104707), CD45.1 (clone: A20; 1:250 dilution, Biolegend,110730), CD45.2 (clone: 104; 1:200 dilution, eBioscience, 11-0454-85), CD49b (clone: DX5; 1:200 dilution, Biolegend, 108904), Fas/CD95 (clone: Jo2; 1:500 dilution, BD Biosciences, 554258), CD138 (clone: 281-2; 1:200 dilution, BD Biosciences, 558626), CXCR4 (clone: L276F12; 1:400 dilution, Biolegend, 146511), CXCR5 (clone: L138D7; 1:50 dilution, Biolegend, 145506), F4/80 (clone: BM8; 1:200 dilution, Biolegend, 123106), GL-7 (1:200 dilution, Biolegend,144610), Gr-1 (clone: RB6-8C5; 1:200 dilution, BD Biosciences, 553124), hCD2 (clone: S5.2; 1:20 dilution, BD Biosciences, 744873), IgD (clone: 11-26c; 1:200 dilution, Biolegend, 405710), IgM (clone: RMM-1; 1:200 dilution, Biolegend, 406512), IL-4Ra (clone: mIL4R-M1; 1:20 dilution, BD Biosciences, 552509), NK1.1 (clone: PK136; 1:200 dilution, Biolegend, 108704), PD-1 (clone: RMP1-30; 1:200 dilution, Biolegend, 109110), and TER-119 (1:200 dilution, Biolegend, 116204). Intracellular staining anti-Ki67 (clone: 16A8; 1:200 dilution, Biolegend, 652410) was described elsewhere[55]. Flow cytometric analysis and cell sorting were performed on a FACS Calibur and a FACSAria III (BD Biosciences) and data were analyzed with Flowjo software (BD Biosciences).

**HA-binding B cells and single-cell analysis**. IgG$^+$ B cells were isolated by the following antibodies: anti-B220 (clone: RA3-6B2; 1:1000 dilution, Biolegend, 103255), CD3 (clone: 17A2; 1:1000 dilution, Biolegend, 100244), CD4 (clone: GK1.5; 1:1000 dilution, Biolegend, 100404), CD5 (clone: 53-7.3; 1:200 dilution, Biolegend, 100604), CD8a (clone: 53-6.7; 1:200 dilution, Biolegend, 100704), CD49b (clone: DX5; 1:200 dilution, Biolegend, 108904), Gr-1 (clone: RB6-8C5; 1:200 dilution, BD Biosciences, 553124), NK1.1 (clone: PK136; 1:200 dilution, Biolegend, 108704), IgD (clone: 11-26c; 1:200 dilution, Biolegend, 405710), IgM (clone: RMM-1; 1:200 dilution, Biolegend, 406512), and TER-119 (1:200 dilution, Biolegend, 116204). HA-binding B cells were identified by APC-conjugated Narita-HA or PE-conjugated PR8-HA that was produced as described elsewhere[56]. The specificity of the HA-binding was validated with APC or PE-conjugated NP$_{29}$-BSA (Supplementary Fig. 11). Analysis and sorting of HA-probe$^+$, B220$^+$, other markers$^-$ cells were carried out with a FACS Aria III. Single B cells culture (Nojima cultures) were performed essentially as described[26,34]. HA-binding B cells were

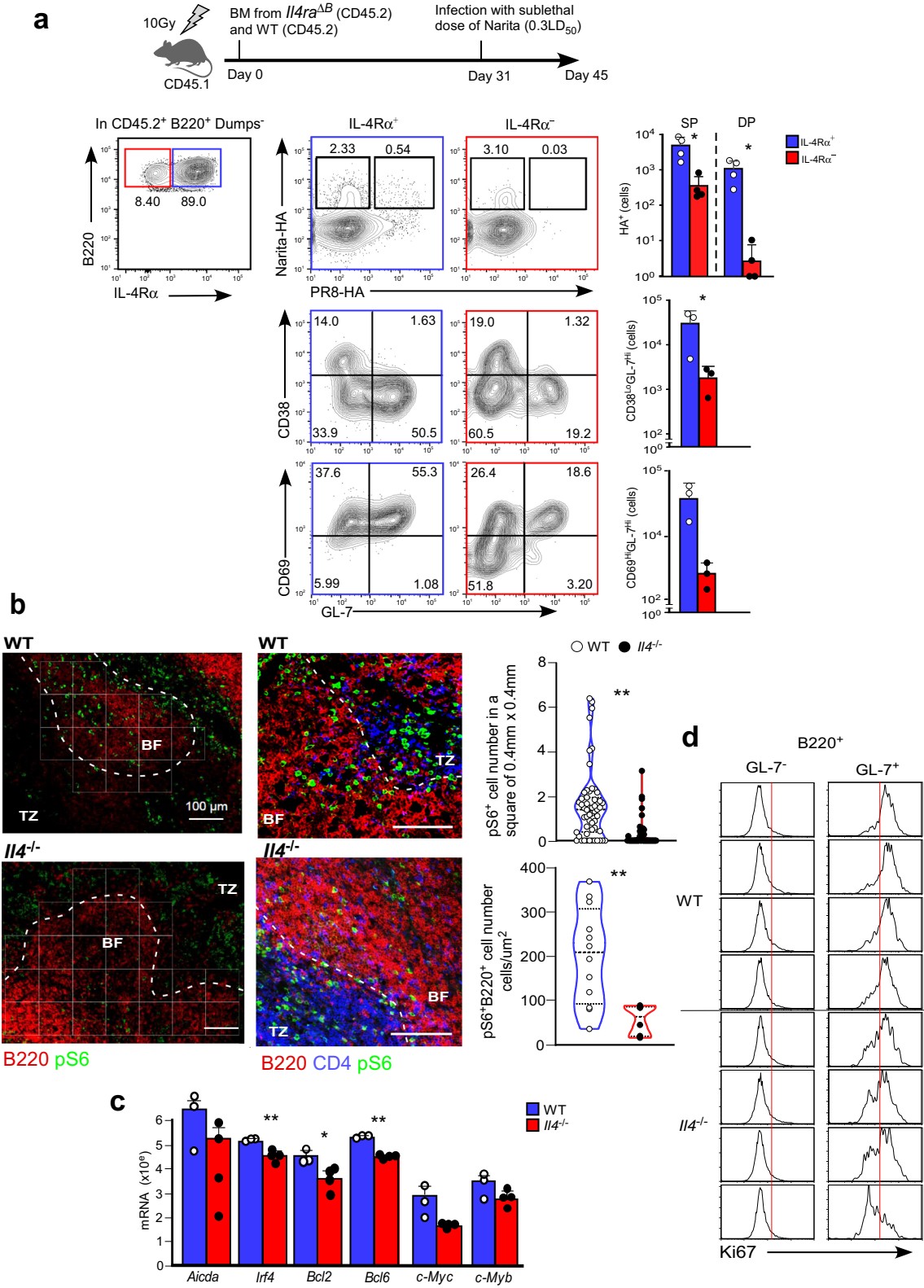

sorted onto the IL-21-expressing CD154+ 40LB fibroblast cell line expressing CD40L and BAFF, NB21.2D9 feeder cells[57] cultured in the presence of recombinant murine IL-4 (Peprotech: 2 ng/ml). Total RNA was isolated with TRIzol (Life Technologies) from the single B cell cultures. RT-PCR was carried out with a SuperScript IV one-step RT-PCR system (Life Technologies) and primers shown in Supplementary Table 1. BCR sequencing data were analyzed with the ImMuno-GeneTics (IMGT) HighV-QUEST (http://www.imgt.org/) in comparison with reference sequences from the IMGT database.

**BCR repertoire analysis**. Total RNA was isolated from freshly sorted Narita-HA binding B cells. cDNA was synthesized using a SMART-Seq v4 Ultra Low Input RNA Kit for Sequencing (Takara Bio, Inc., Shiga, Japan, Z4888N). Ig sequences were amplified by PCR using KOD Fx Neo (Toyobo, Osaka, Japan) and primers shown in Supplementary Table 1. cDNA libraries were synthesized using a Nextera DNA Library Prep Kit (Illumina, San Diego, CA, FC-131-1096) and sequencing data were obtained using the MiSeq system (Illumina). Sequenced reads were analyzed with the IMGT HighV-QUEST (http://www.imgt.org/) after removing

**Fig. 7 IL-4 signaling in B cells is crucial for the metabolic shift during GC B development. a** Top panel indicates the design of the BMT experiment. Irradiated CD45.1 host mice were reconstituted with a mixture of BM cells from WT (blue bars) and $Il4ra^{\Delta B}$ CD45.2 mice (red bars) ($n = 3$ for each). The host mice were infected with A/Narita/1/2009 on day 31. Fourteen days later, Binding to Narita-HA and/or PR8-HA and generation of CD38$^-$GL-7$^+$, or CD69$^+$GL-7$^+$ cells in the B220$^+$ B cells, or CD38$^{dull}$Fas$^+$ cells in the CD19$^+$Dump$^-$ populations, (right graph) were compared between IL-4R$^+$ and IL-4R$^-$ B220$^+$ B cells. **b** Naive OT-II T cells from WT or $Il4^{-/-}$ mice were transferred into $Cd28^{-/-}$ mice followed by immunization with OVA + alum. Frozen sections were prepared at 6 days post-immunization and stained with antibodies against B220, pS6, and CD4 (left 4 panels). Left panels illustrate the localization of pS6$^+$ B cells in the spleen. Dashed lines demarcate the follicle area, and these areas are divided into 0.01 mm$^2$ squares (white lines). Each dot in the violin plot (right top panel) represents the Number of pS6$^+$ B cells localized in each 4 mm$^2$ square areas. 154 and 99 sites were counted in the WT (white dots) and the $Il4^{-/-}$ (black dots) section, respectively. The bar graph in the bottom right panel indicates the mean number of pS6$^+$ B cells per μm$^2$. Scale bar indicates 100 μm. **c** mRNA expression levels were measured in splenic B cells from WT (blue bars, $n = 3$) or $Il4^{-/-}$ (red bars, $n = 4$) T cell-transferred mice. **d** Ki67 expression was analyzed in splenic B220$^+$ B cells from the transferred mice from WT ($n = 4$) or $Il4^{-/-}$ ($n = 4$) mice. Ki67 expression is analyzed in GL-7$^+$ (right) and GL-7$^-$ (left) B cells. Bars represent mean; *$p < 0.05$, **$p < 0.01$ by ratio paired, two-tailed $t$-test (IL-4Rα$^+$ group with IL-4Rα$^-$ in **a**), or by unpaired, two-tailed $t$-test (WT group with $Il4^{-/-}$ group in **b** and **c**). All error bars represent SEM.

low-quality reads, trimming of adapter sequence, and connecting read1 and read2 using tools available on the Galaxy platform (https://usegalaxy.org).

**Statistics and reproducibility**. Statistical comparisons between groups were performed using Prism version 8.0.2 (Graph Pad Software, San Diego, CA). Data represent the mean ± SEM. Two groups were compared using the two-tailed, unpaired Student's $t$ test. All experiments were performed at least twice, and the similar results were obtained.

**Reporting summary**. Further information on research design is available in the Nature Research Reporting Summary linked to this article.

## Data availability

The BCR sequencing data have been deposited in GEO under the accession code GSE168976. Microscopy image data are available from the corresponding author upon reasonable request. Source data are provided with this paper.

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

## Acknowledgements

We thank Dr. Manfred K. for providing *Il4*−/− *and Il6*−/− mice. We thank Dr. Brombacher F. for providing *Il4ra*−/− and *Il4ra*f/f mice. *Tmprss2*−/− mice, *Gata3*f/f ert2cre mice, and Verigem mice were kindly gifted from Dr. Takeda, M., Dr. Zhu, J., and Dr. Allen C. We thank Dr. T. Kitami, S. Ki and Y. Suzuki, for technical support and animal maintenance. We thank Dr. P. Burrows for his helpful comments on the manuscript. This work was supported by a Grant-in-Aid for Scientific Research (B) (19H03491) to M.K. Scientific Research (C) (18K06647) to K.M. JSPS KAKENHI (S) (JP17H06175) and AMED-CREST (JP20GM1110009) grants to A.Y. JSPS KAKENHI (JP19K17656) and AMED (JP20JK0108141) grants to Y.T. This work was supported by Mochida Memorial Foundation for Medical and Pharmaceutical Research, and Daiichi-Sankyo Foundation of Life Science.

## Author contributions

M.K. designed and conceptualized the research and K.M. performed mouse infections and T.Y. and Y.H. performed OVA system for BCR affinity experiments and immunohistostaining Y.A., K.T., Y.T., O.O., and K.M. performed Ig sequencing analysis; H.H. established H1N1 Narita strain; H.F. established mouse-adapted H2N2 strain; Y.A., and Y.T. established and provided H1N1 HA probes; Y.I., A.Y., and S.F. distributed mice; S.D. and K.Y. performed pathway enrichment analysis; K.M. and M.K. prepared the manuscript.

## Competing interests

The authors declare no competing interests.
