## [Peer Review File · Nature Communications]

REVIEWER COMMENTS

Reviewer #1 B cells, Tfh, influenza (Remarks to the Author):

In this manuscript Miyauchi et al. explore the mechanisms underlying the observation that heterosubtypic immunity to influenza was induced after infection, but not after intraperitoneal injection of inactivated virus in alum. The topic, exploring questions of heterosubtypic immunity induction is of significance, given the yearly changing influenza viruses and the current focus on identifying ways to induce broadly-protective antibodies to enhance vaccine efficacy. Overall, they conclude from their studies that TFH-driven IL-4 production is required to drive a robust clonal expansion of germinal centers (GC) after infection that generate B cell responses with heterosubtypic binding, as in the absence of CD4-derived IL-4 or B cells lacking the IL-4Ra GC are reduced and dual-binding B cells are reduced also. It is puzzling that no data are provided to show whether vaccination fails to induce IL-4 producing TFH, or whether vaccination-induced responses are affected in the absence of IL-4 production. Thus, in the end the paper does not come back completely to the question it began to address.

The manuscript contains an extensive (and exhaustive) number of experiments and figures, many of high quality, some require clarifications. In the first part of the manuscript they conduct numerous lavage and serum transfer studies from mice either infected or vaccinated, to demonstrate induction of protective capacity of these fluids after infection when intranasally induced before homo- or heterosubtypic viral challenge. Protection was shown to depend on robust viral replication and was lacking when fluids were passed over IgG and IgM-binding Protein G. Using fluorescent-baits for disparate HA's they discovered B cells able to bind to H1 of A/PR8 and H1 of a 2009 pandemic train of influenza H1, but only in mice infected, not mice vaccinated ip. They then sorted total HA-binding cells at the single cell level and conducted Ig-heavy chain sequencing comparing splenic B cells after vaccination with lymph node B cells after infection, showing repertoire differences. Next, they used mice lacking in either CD4 or B cells expressing Bcl6 to indicate the importance of germinal centers in development of cross-protective immunity 14 days after infection (Bcl6 was needed for induction of protection), then used RNA sequencing on a number of myeloid and epithelial cells from the lung to deduce inflammatory signals that might be regulating induction of cross-protection. IL-6 had some effect. They then returned to study IL-4, IL-21 and IFN-g, all products of TFH on their effects on the GC responses. Using both influenza and then the OT-II transgenic system to demonstrate in a number of different ways the need for B cell IL-4 signaling and IL-4 production by CD4 T cells in the induction of GC B cells.

The overall conclusions from the paper are thus that robust GC responses are beneficial for induction of protection and that IL-4 drives GC responses, neither finding novel. One is left wondering whether the ip injection of virus in alum is simply not strong enough to induce a sufficiently long-lasting GC response. Thus, whether this is all related to the magnitude and duration of the response. Studies to compare the effectiveness of antibodies adjusted for their concentration would be important to address this point (quality versus quantity). Another aspect that is puzzling is the timing of the influenza response. IgG responses on day 14 after influenza infection would be derived mainly, if not exclusively, from extrafollicular responses. GC do not usually begin to emerge until around day 10 or so after infection, i.e. close to the time of virus clearance. It seems incongruent that the protective capacity generated by secreted antibodies lies with the GC response, which generates memory early rather than plasma cells, a fact not discussed.

There are numerous parts in the manuscript that could benefit from clearer explanations, and a better referencing of the literature, especially when it comes to discussing the B cell response to influenza.

Specific Comments:

1) Figure 1 Infection with influenza virus in mice induces robust IgM responses in the serum as well as locally in secretions. The statements and figure should be amended to acknowledge that fact. Serum/BALF from infected B-deficient mice would be a much cleaner system, but at least data interpretation should take IgM into consideration. It is also unclear how the authors distinguished upper from lower lung (nasal infection is clear) respiratory tract production of

antibodies. Seems very difficult by ELISA. Finally, the authors should acknowledge previous work demonstrating that IgG is the main isotype produced in mouse lungs in response to influenza (all the way back to 1987 if not earlier– Jones et al.). These studies were done by ELISPOT, which seem easier to interpret than the data shown here. Reference would suffice.

2) Figure 1C does not show that infection induced high levels IgG and IgA – vaccination did. Is this a mix-up in the labeling? Why as there no antibody induced by the vaccination (or infection – depending on the label). This seems surprising. Thus, could the difference observed be simply a difference in the concentrations of antibody produced? Have the authors considered differentiating quality from quantity by adjusting concentrations of influenza-specific antibodies. If the response to vaccination is just lower -that would be a trivial explanation for their findings, and should be excluded. This interpretation would also be supported by their finding in Figure 2 that the (much lower) levels of responses in the *Tmprss2*^{-/-} mice are protective, but only to the homologous strain – if the levels of IgG were adjusted to that of the wt – would the difference to A/PR8 disappear?

3) Why do the authors refer to IgG2 – is that IgG2c or IgG2b? Two very different IgG molecules (Figure 2) with very different effector functions.

4) The fluorescent-bait staining requires proper controls with baits of the same color but unrelated specificity. FMO controls must be shown for each bait.

5) The analysis of the repertoire (Fig. 3) is unclear. In the M&M it states that lung was isolated for RNAseq. The Figure legend states that vaccinated mouse B cells were taken from the spleen and infected from the draining lymph nodes. But it is unclear whether these cells were pooled or from individual mice (which is critical for repertoire analysis and comparison for the dual versus single-specific B cells for example). Given the differences in the repertoire of individual mice the data are insufficient to determine whether any sequence differences were due to the difference in individual mice, different tissue locations, and or mode of infection/vaccination.

6) Figure 5. This might be a misunderstanding, but how can genes associated with TCR differentiation and BCR signaling be induced in cells like eosinophils and other myeloid and epithelial cells? Were these contaminations in their cell preps. This is very confusing.

7) The authors do not demonstrate whether immunization in alum fails to induce IL-4 by TFH to explain the first part of their data.

8) Figure 6, the difference in “dual” recognizing B cells appear to be 50 cells (350 versus 300). If that is correct- it seems a fairly minor difference to explain a strong difference in protection.

Minor Comments:

1) Line 126-127. It is unclear what the authors mean by “However, these antibodies recognized a 30% difference” – why “however”?

2) The vaccination strategy is not fully outlined in the M&M section. What was the timing of the 2 injections?

3) When was virus-clearance achieved after primary infection for each of the strains at the time of challenge?

4) How did the author reach conclusions about the influenza-specific antibody concentrations? – A influenza and isotype-specific standard must be used and was not indicated in the M&M section.

5) Figure 2A states that viral load was determined on day 10 after infection, but all wt mice are dead by day 6? Please clarify. Also, the authors should measure virus load at their peak (days 2-3 after infection) to get a better measure of the differences in viral replication.

6) Line 201 – it is well understood by the field that the draining lymph node and not the spleen harbor the effector B cell responses after influenza infection. There are dozens of papers the authors should consider citing that have already made these conclusions.

7) Figure 3e. The Y-axis is unclear – how did they determine 100% maximum binding? Is this based on antibody concentration or binding strengths, or both?

8) Line 225, LAH needs reference

9) Line 273, the conclusion that GC responses occur in the lung is not supported by their data, which show GC responses in the draining lymph node. The statement must be amended.

10) Line 351, “IL-4 deficiency in B cells”....? This is not shown

Reviewer #2 influenza vaccines, Tfh (Remarks to the Author):

The manuscript by Miyauchi et al explores the role of IL-4 in supporting germinal B cell function following infection or immunization. There is a tremendous amount of work presented, however the overall study suffers from a lack of clarity in the execution, analysis and rationale for the experiments carried out. While many of the models and techniques are technologically advanced, the limited explanation and justification for many of the experimental choices, combined with limited details about the reproducibility of the observations, make the strength of the authors observations very difficult to tease out. I suggest significantly reducing the scope of work presented to make the message more cogent.

Major comments –

1 – For all experiments, the animals numbers per group and number of times each experiment was independently repeated is very unclear. For some panels, 2-3 animals are used while for others consist of groups of 7 or more. What is the justification for the different animal groups sizes? Similarly why are t-tests used for group comparison in some figures, but Mann whitney U tests used for later experiments. What is the justification for selection of these tests and why change? Also, moving back and forth of the flow data from cell frequencies to absolute numbers presented in the column graphs is confusing, and total cell counts in each sample should be presented also in these instances.

2 - Figure 1 shows cross-protection of mice against PR8 if pre-infected with pH1N1, but protection was not observed with prior immunization. Protection against H2N2 is not very convincing. Given the premise of this paper is that the quality of the antibody responses (specifically the degree of cross-reactive specificities) can confer heterologous protection, it is puzzling why the authors did not directly quantitate antibody against HA, NA, NP and the HA stem for both pdmH1N1 and PR8 (and H2N2) from their immunized and pre-infected animals.

3 – Figure 1 - The serum and BALF transfer experiments are not that informative. These should be harvested from both vaccinated and pre-infected animals, and then the amount of antibody transferred equalized based on total amounts of anti-Narita binding Ig (or titres of anti-Narita antibody in serum at least). This would establish that qualitative differences in the antibody response, and not the magnitude of the responses underpin the differential protection observed against PR8 challenge.

4 – Figure 1 – differential infusion volumes were used to control the depth of inocula (5ul for upper only, 50 for upper and lower). However, there was no concentration of virus in the mice administered 5ul, and the mice administered 50 showed consistently more virus in both upper and lower lung. The conclusion that lower lung is responsible for the antibody is not clear given more replication overall with the larger volume.

5 – Figure 2 seems redundant. Basically mice that cannot sustain infection, don't generate antibodies, that then don't protect against challenge. Similarly to Sup Fig2. Note the WT controls offer very little protection in this experiment, although the dosing indicated is now given as pfu instead of LD50 in the first figure so is difficult to directly compare (and not indicated at all on Supp 2).

6 - Figure 3A. Based on figure legend descriptors, the data shows splenic-derived B cell staining for the vaccinated animals via IP route. Do mice, instilled with inactivated virus via intranasal route, induce similar mono-specific Narita+ B cells in the MLN? There is a possibility that TMPRSS2 modifies the HA of the inactivated virus, despite the absence of replication, giving rise to cross-reactive epitopes. The data presented may indicate effects imparted by route of administration (IN vs IP) rather than the requisite for viral replication.

7 – Figure 3 – It is unclear why the comparison is spleen (for vaccinated animals) and MLN (for infected), there could be anatomical differences that alter staining patterns of the HA probes. Both organs from both experimental conditions should be shown and analysed similarly by BCR sequencing and/or monoclonal generation. Also, it is unclear how many animals were analysed in

figure 1a, please use dot plots instead of bar charts. Also, upstream flow cytometry gating trees need to be provided for each tissue.

8 – Figure 3 - The BCR sequence analysis doesn't specify how many clones were recovered, how many animals were sorted? How variable was the repertoire between animals? How variable is the repertoire within an animal but between anatomical sites? Without this information the data in fig3c and 3d and 3e don't have any meaning.

9 – Figure 3 - How many mice were the Nojima culture antibodies derived from? Many studies have shown mice can make stem antibodies to infection or vaccination (Angeletti et al 2017, Tan et al 2019 Angeletti et al 2019) and can participate in GC reactions. The significance of the SHM data is unclear, especially given the very small number of clones analyzed. In either case, dual-specific or single-specific derived clones have very similar mutation loads. The authors assert that both "SHM of their immunoglobulin 241(Ig) genes is likely to be essential" and "SHM is not always necessary" for the generation of anti-stem antibodies. This is well established in the literature but not well supported by the included experiments in this study.

10 – Figure 4 - BCL6 loss in T or B cells compromises the generation of a robust antibody response as expected. However, again there was no equalization of amounts of binding antibody in the BALF for the transfer experiment, so magnitude and quality changes in the serological response cannot be split apart.

11 – Figure 4 - The probe binding populations should be assessed on both the GC B cells (GL7+ or similar) and the bulk IgD- population. Are the cross-reactive B cells in the GC or not?

12 – Figure 6 – It is interesting that IL-21 KO mice were able to maintain WT levels of GC activity. This seems contrary to the essential role IL-21 plays in GC formation, and implies that non-T cell sources of IL-21 can substitute for Tfh. Do the authors have an explanation for this?

13 – Figure 6. While the data shows anti-PR8 immunity is affected and the dual-probe population is reduced, the lack of protection may be rather a consequence of significant total GC B reduction. To show that the dual-probe population is indeed affected by IL4 deficiency, and not the Narita+ GC B cells, experiments could be repeated with a Narita challenge model as well to show that IL4-deficiency still retains homologous protection.

14 – Figure 7 - The change to OTII / OVA immunization models adds a further level of complication without addressing the initial premise. The authors should just use the bone marrow chimeras of the IL4Ra B cell KOs following Narita infection, you would expect only IL4Ra competent B cells to contain a dual probe population, while the KO B cells should show Narita probe staining only. That would help to address the total GC defect in the IL4Ra KO mice.

Minor comments:

1 – "these human anti-stalk mAbs recognize conserved regions of the HA among group 1 and group 2 influenza A viruses, and the antibody heavy-chain variable region genes are heavily mutated." - This is not supported in the literature and seem perfectly normal level of mutation for antigen experienced B cells.

2 - All flow plots should be moved off the axis, hard to clearly see the gated populations.

3 – All column graphs should have individual data points.

Reviewer #3 antibody, vaccines (Remarks to the Author):

In this study, the authors performed a series of experiments on anti-influenza HA antibody responses to an inactivated vaccine or a wild type virus infection, focusing cross-reactive immunity using animal models (with mutant mice). The authors' major claims are that a live virus infection, through virus replication in the lung, induces more cross-reactive (broad) immunity which requires Tfh and germinal centre response, and IL-4 signalling is critical in the expansion of GC B cells recognising cross-reactive epitopes, thus important in activating broad anti-influenza (HA)

antibody immunity.

Although it is already known a live infection would normally activate broader immunity and memory than an inactivated or subunit vaccine, this study provides important new insights on how a broader, cross-reactive antibody response to influenza (flu) HA is activated and demonstrates a critical role for Tfh-mediated IL-4 signalling in enhancing the broad anti-flu antibody responses targeting conserved HA epitopes and immune memory. This may have implications in new vaccination strategies such as the development of "universal" flu vaccines.

The authors may find it helpful to add/modify/address the following points.

1. On the implications from the study findings, it would be helpful to add a bit more discussion on how to use this knowledge to improve current vaccine or therapeutic strategy.
2. On the use of different virus infection titres, i.e. 5ul or 50ul, to define upper or lower respiratory tract infection, is there any evidence or previous literature about this approach? No details were given on this aspect in the paper. In Figure 1e, it isn't clear to me the meanings of "Lung upper" and "Lung lower".
Also, it may be helpful to explain a bit more on how to determine where the antibodies are produced/generated, i.e. upper or lower respiratory tract.
3. Individual methods (methods section) seem rather brief, which may be limited by word limit.

Point-by-point response

Reviewer #1 B cells, Tfh, influenza (Remarks to the Author):

In this manuscript Miyauchi et al. explore the mechanisms underlying the observation that heterosubtypic immunity to influenza was induced after infection, but not after intraperitoneal injection of inactivated virus in alum. The topic, exploring questions of heterosubtypic immunity induction is of significance, given the yearly changing influenza viruses and the current focus on identifying ways to induce broadly-protective antibodies to enhance vaccine efficacy. Overall, they conclude from their studies that TFH-driven IL-4 production is required to drive a robust clonal expansion of germinal centers (GC) after infection that generate B cell responses with heterosubtypic binding, as in the absence of CD4-derived IL-4 or B cells lacking the IL-4Ra GC are reduced and dual-binding B cells are reduced also. It is puzzling that no data are provided to show whether vaccination fails to induce IL-4 producing TFH, or whether vaccination-induced responses are affected in the absence of IL-4 production. Thus, in the end the paper does not come back completely to the question it began to address.

I apologize for the disorganization and the lack of a clear and logical explanation. In this manuscript, in the present manuscript, the following two points are the main focus.

1: The production of broadly reactive antibodies requires virus entry through the respiratory tract and viral replication in the deep lung, which induces a structural change in the virus HA that exposes rare antigenic epitopes. These rare epitopes are needed to activate a minor B cell population that produces broadly reactive antibodies.

2: As a result of virus infection, this minor B population differentiates and produces broadly reactive antibodies in the MLN. The T_{FH}-derived IL-4 is to induce GC formation in which the B cell repertoire of broadly reactive antibodies, which exist only at low frequency, is efficiently expanded.

Regarding point 2, we have previously shown that T_{FH} derived IL-4 can be found even after vaccination with inactivated Narita virus (Miyauchi, NI, 2016), but the broadly reactive antibody cannot be detected in this case. Therefore, the

lack of broadly reactive antibodies after vaccination cannot be explained by the absence of IL-4 signaling. The present study demonstrated that the importance of the T_{FH}-derived IL-4 is to induce GC formation in which the B cell repertoire of broadly reactive antibodies, which are present only at low frequency, is efficiently expanded. The broadly reactive antibodies recognize the antigen epitopes on HA that normally do not appear on the surface. This epitope only appears after the conformational change in HA that occurs during virus infection. Therefore, the main role of T_{FH} derived IL-4 is to efficiently expand B cells in the GC that recognize this rare epitope. We believe that failure to induce the broadly reactive antibodies by the inactivated vaccine is because of the absence of this rare epitope. We summarized this issue in the first paragraph (L321 to L336) and the sentence between L417 to L426 of discussion.

The manuscript contains an extensive (and exhaustive) number of experiments and figures, many of high quality, some require clarifications. In the first part of the manuscript they conduct numerous lavage and serum transfer studies from mice either infected or vaccinated, to demonstrate induction of protective capacity of these fluids after infection when intranasally induced before homo- or heterosubtypic viral challenge. Protection was shown to depend on robust viral replication and was lacking when fluids were passed over IgG and IgM-binding Protein G. Using fluorescent-baits for disparate HA's they discovered B cells able to bind to H1 of A/PR8 and H1 of a 2009 pandemic strain of influenza H1, but only in mice infected, not mice vaccinated ip. They then sorted total HA-binding cells at the single cell level and conducted Ig-heavy chain sequencing comparing splenic B cells after vaccination with lymph node B cells after infection, showing repertoire differences. Next, they used mice lacking in either CD4 or B cells expressing Bcl6 to indicate the importance of germinal centers in development of cross-protective immunity 14 days after infection (Bcl6 was needed for induction of protection), then used RNA sequencing on a number of myeloid and epithelial cells from the lung to deduce inflammatory signals that might be regulating induction of cross-protection. IL-6 had some effect. They then returned to study IL-4, IL-21 and IFN- γ , all products of TFH on their effects on the GC responses. Using both influenza and then the OT-II transgenic system to demonstrate in a number of different ways the need for B cell IL-4 signaling and IL-4 production by CD4 T cells in the induction of GC B cells.

The overall conclusions from the paper are thus that robust GC responses are beneficial

for induction of protection and that IL-4 drives GC responses, neither finding novel. One is left wondering whether the ip injection of virus in alum is simply not strong enough to induce a sufficiently long-lasting GC response. Thus, whether this is all related to the magnitude and duration of the response. Studies to compare the effectiveness of antibodies adjusted for their concentration would be important to address this point (quality versus quantity).

We apologize for the confusing description. Our main point about the GC responses during virus infection is their beneficial role for expansion of the rare B cells responsible for the broadly reactive immunity. We believe this is a novel finding but that our discussion did not make this point clear. The GC responses are also induced by vaccination (Miyachi, NI, 2016) and thus the difference between the systemic response induced by the inactivated vaccine and the local response induced by the nasal infection cannot be simply explained by the intensity of GC responses. Therefore, a reasonable explanation is that the difference between systemic vaccination and local infection is due to differences in the repertoires of the responding B cells. Our results of the B cell binding to heterotypic HA (revised **Fig.3a**) and BCR repertoires (revised **Fig.3c-d**) support this idea. To clarify the difference in the B cell repertoires between vaccination and the nasal infection, we revised the sentence of L173 to L190.

Another aspect that is puzzling is the timing of the influenza response. IgG responses on day 14 after influenza infection would be derived mainly, if not exclusively, from extrafollicular responses. GC do not usually begin to emerge until around day 10 or so after infection, i.e. close to the time of virus clearance. It seems incongruent that the protective capacity generated by secreted antibodies lies with the GC response, which generates memory early rather than plasma cells, a fact not discussed.

We thank the reviewer for this important comment. Our data strongly support the dependency of broadly-reactive antibody responses on the GC response. We believe the main role of the GC response might be an expansion of B cells rather than memory generation in the case of the generation of broadly-reactive antibody responses. We found CD38⁺ B cells in the MLN at 10-day post-infection and the GC response was exclusive with plasma cell response (revised **S-Fig.5**). Moreover, infection conferred protective responses at 40 days after infection (revised **S-Fig.6**), indicating that the GC response is required for

the B cell memory response (L245-L248). However, our BCR sequence data suggested that GC responses against heterosubtypic HA antigen may have different roles other than SHM and affinity maturation of antibodies (**Fig.3f&g**).

We described these issues in the sentence of L202-L216 and L262-L269.

There are numerous parts in the manuscript that could benefit from clearer explanations, and a better referencing of the literature, especially when it comes to discussing the B cell response to influenza.

Thank you for the valuable suggestion. We revised the descriptions of anti-influenza B cell responses in the Discussion, with appended references (Victra GD et al 2015 Cell, Sangesland M et al 2019 Immunity, Angeletti D et al 2019 PNAS, Adachi Y et al 2019 Nat. Commun, Krammer F et al Nat Rev Immunol 2019) to clearly explain possible mechanisms in viral replication inducing broadly protective antibody responses.

Specific Comments:

1) Figure 1 Infection with influenza virus in mice induces robust IgM responses in the serum as well as locally in secretions. The statements and Figure should be amended to acknowledge that fact. Serum/BALF from infected B-deficient mice would be a much cleaner system, but at least data interpretation should take IgM into consideration.

We agreed with the reviewer's comment. To eliminate the contribution of IgM antibodies, we removed the IgM from the BALF by passing it through an anti-IgM antibody-coupled Sepharose 4B column. Since the protective activity remains intact even in the BALF depleted of IgM Abs, we conclude that the contribution of IgM is much less than IgG. We added new data to revised **Fig.2a** and **S-Fig.2**, and the sentence between L145 and L149.

It is also unclear how the authors distinguished upper from lower lung (nasal infection is clear) respiratory tract production of antibodies. Seems very difficult by ELISA. Finally, the authors should acknowledge previous work demonstrating that IgG is the main isotype produced in mouse lungs in response to influenza (all the way back to 1987 if not earlier– Jones et al.). These studies were done by ELISPOT, which seem easier to interpret than the data shown here. Reference would suffice.

Thank you for the useful suggestion. We used an ELISPOT assay for mediastinal lymph nodes (MLN) and spleen of the infected mice. On day 14 post-infection, the MLN showed a high number of IgG⁺ ASC in response to influenza HA, and revised **Fig.3b** showed a higher number of dual-specific IgG B cells than those in the spleen. Therefore, we concluded that the MLN was the major site for the production of anti-HA antibodies. We added these data to **Fig.2c**. We also acknowledged previous studies by Ada's group (Jones PD et al 1987 Vaccine).

2) Figure 1C does not show that infection induced high levels IgG and IgA – vaccination did. Is this a mix-up in the labeling? Why as there no antibody induced by the vaccination (or infection – depending on the label). This seems surprising. Thus, could the difference observed be simply a difference in the concentrations of antibody produced? Have the authors considered differentiating quality from quantity by adjusting concentrations of influenza-specific antibodies. If the response to vaccination is just lower – that would be a trivial explanation for their findings, and should be excluded. This interpretation would also be supported by their finding in Figure 2 that the (much lower) levels of responses in the *Tmprss2*^{-/-} mice are protective, but only to the homologous strain – if the levels of IgG were adjusted to that of the wt – would the difference to A/PR8 disappear?

As reviewer #1 pointed out, the labeling of Fig.1c was a mistake. We have revised **Fig.1c** and the explanation in L138-L143.

Regarding second issue, our data indicated that systemic IgG responses against the vaccinated virus Narita HA showed the equivariant or higher titers in the serum as in the infection. On the other hand, the IgG response to PR8, a seasonal virus strain, was not detectable in the vaccinated mice. As this reviewer points out, the reduction in IgG titer against PR8-HA was unexpected because there should be sheared epitopes between these two strains. However, the issue that we address here is that vaccination and infection induce qualitatively different antibody responses in the protective responses against heterotypic viruses. To emphasize our point, we used serum from the infected mice and compared the protective responses in the same anti-Narita-HA titer between vaccination and infection (revised **Fig.1a-c**).

The difference in the IgG response between WT and *Tmprss2*^{-/-} mice is

observed only when live virus is administered intranasally, as in a natural infection. This is not a vaccination experiment. We apologize for the confusing explanation. Since the influenza virus cannot replicate in *Tmprss2*^{-/-} mice, the mice fail to generate the IgG responses even after nasal administration with a thousand times more virus (3000LD₅₀) than the sublethal dose (0.3LD₅₀) for WT mice. However, this high dose administration is enough to induce HA-specific serum IgG in *Tmprss2*^{-/-} mice (revised **Fig.4b**). Therefore, this system allowed us to evaluate the protective ability of IgG antibodies from *Tmprss2*^{-/-} mice against PR8 infection. In the previous experiment, we had an inaccurate quantitative measurement of the protective ability against PR8. Therefore, we added a new experiment to measure the protective ability in which we adjusted the serum concentration based on the IgG titer against Narita (revised **Fig.4c**). This experiment indicates that intranasal administration with a sufficient amount of antigen can induce neutralizing antibodies against the specific antigen Narita, but not broadly reactive IgG antibodies with neutralizing ability against PR8. This result strongly supports our explanation that the replication process of the virus through the route of infection is important for efficiently generating broadly reactive IgG antibodies. We described these issues in the sentence of L218 to L243.

3) Why do the authors refer to IgG2 – is that IgG2c or IgG2b? Two very different IgG molecules (Figure 2) with very different effector functions.

We thank you for pointing this out. Considering the characteristics of the isotype, we have standardized on IgG2b, which has high ADCC activity. We revised **Fig. 4b** and **5b** accordingly.

4) The fluorescent-bait staining requires proper controls with baits of the same color but unrelated specificity. FMO controls must be shown for each bait.

We agree with this comment about the specificity. To further verify the specificity of HA probes, we additionally tested whether B cells obtained from Narita virus-infected mice bind to the NP₂₉-BSA probes, according to the suggestion from reviewer #1. The results indicate that our system accurately demonstrates binding to the virus-derived HA antigen. We added revised **S-Fig.11** and the explanation in L517 to L519 in Materials and Methods.

5) The analysis of the repertoire (Fig. 3) is unclear. In the M&M it states that lung was isolated for RNAseq. The Figure legend states that vaccinated mouse B cells were taken from the spleen and infected from the draining lymph nodes. But it is unclear whether these cells were pooled or from individual mice (which is critical for repertoire analysis and comparison for the dual versus single-specific B cells for example). Given the differences in the repertoire of individual mice the data are insufficient to determine whether any sequence differences were due to the difference in individual mice, different tissue locations, and or mode of infection/vaccination.

As this reviewer points out, we used the splenic B cells for vaccination and the MLN for nasal administration. In this case, we compare two different secondary lymphoid origins responsible for the antibody response, the spleen and the MLN. As shown in **Fig.3a**, virus infection via nasal administration induces the antibody response in both the spleen and dLN, but dual-specific B cells were rare in the spleen. On the other hand, vaccination intraperitoneal administration is not sufficient to activate the B cell response in the MLN (L182-L184). Therefore, the comparison of the B cell repertoire between the spleen from vaccinated mice and the MLN from infected mice is one way to examine the difference in the IgG responses. In the original manuscript, we have shown representative data from one mouse per immunization. In the revised **Fig.3d**, we analyzed three additional mice and show the results of individual mice. We revised **Fig.3a** and **Fig. 3d**, and the sentence between L184-L190

6) Figure 5. This might be a misunderstanding, but how can genes associated with TCR differentiation and BCR signaling be induced in cells like eosinophils and other myeloid and epithelial cells? Were these contaminations in their cell preps. This is very confusing.

We agree with this comment about the previous **Fig.5**. This Fig. focused on the genes related to TCR or BCR signal pathways, but their expression is not limited to T or B cells. However, as this reviewer points out, this issue is irrelevant to the main argument of this paper. The editor asks that the manuscript should be simplified, we have removed these RNA seq data in the revised version.

7) The authors do not demonstrate whether immunization in alum fails to induce IL-4 by TFH to explain the first part of their data.

We have previously demonstrated that IL-4 production did occur after vaccination with inactivated virus (Miyachi K et al 2016). However, vaccination fails to generate broadly-reactive IgG antibodies. Therefore, the production of the broadly-reactive IgG antibodies in MLN needs something else besides the presence of IL-4. This something else only occurs during actual live virus infection. The generation of broadly reactive antibodies requires structural changes in the virus HA that expose common antigenic epitopes that do not normally appear on the surface. We believe that the role of IL-4 is to efficiently expand the B cells that respond to this rare sheared antigen.

8) Figure 6, the difference in “dual” recognizing B cells appear to be 50 cells (350 versus 300). If that is correct- it seems a fairly minor difference to explain a strong difference in protection.

We apologize for the unclear notation. The Y-axis is a logarithmic scale. We modified this in revised **Fig.5c** to make it more clear.

Minor Comments:

1) Line 126-127. It is unclear what the authors mean by “However, these antibodies recognized a 30% difference” – why “however”?

According to the reviewer’s suggestion, we removed “However” and the sentence make clearer at L119-L121.

2) The vaccination strategy is not fully outlined in the M&M section. What was the timing of the 2 injections?

We apologize for the incomplete descriptions in the Materials and Methods section. Mice were immunized twice with an interval of 7 days. We revised the Materials and Methods section on L453-L459.

3) When was virus-clearance achieved after primary infection for each of the strains at the time of challenge?

The virus-clearance was achieved until 13 days after primary infection of a sublethal dose of Narita and PR8. The virus introduced in the initial infection is almost eliminated by the time of the second infection as a virus challenge. To clarify this, we measured viral load instead of virus RNA and revised **Fig.2b**.

4) How did the author reach conclusions about the influenza-specific antibody concentrations? – A influenza and isotype-specific standard must be used and was not indicated in the M&M section.

I agree with the point raised by this reviewer. In the original version, we calculated the influenza-specific antibody concentrations based on the ratio to the known antibody concentrations. According to this reviewer's suggestion, we establish the isotype-specific standard with recombinant IgG2b and IgA antibodies reactive with the H1N1 HA based on the IgG1 (Adachi Y et al 2015 JEM) and IgG2c (Watanabe A et al 2019 Cell) monoclonal antibodies. We reevaluated the antibody concentrations and revised **Fig.1c, 2b&d, 4b, 5b, 6a**. We also changed the Materials and Methods section on L484-L488.

5) Figure 2A states that viral load was determined on day 10 after infection, but all wt mice are dead by day 6? Please clarify. Also, the authors should measure virus load at their peak (days 2-3 after infection) to get a better measure of the differences in viral replication.

I apologize for the confusing description. Yes, all WT mice were euthanized by day 8 in **Fig.4a**, because body weight fell below 75% of the starting weight. We measured the serum titer of the IgG antibodies in the mice infected with a sublethal dose, 0.3LD₅₀. We added the body weight change of WT mice at a lethal dose and the sentence "The mice whose body weight fell below 75% of the starting weight were euthanized" in the Figure legend of . We use the same format in the Figure indicating the body weight change.

Furthermore, as the reviewer pointed out, we added the experiment to measure the virus titer of *Tmprss2*^{-/-} mice at day 3 post-infection, and we added new data in **Fig.4a** to confirm no viral replication on both days 3 and 10.

6) Line 201 – it is well understood by the field that the draining lymph node and not the

spleen harbor the effector B cell responses after influenza infection. There are dozens of papers the authors should consider citing that have already made these conclusions.

We appreciate the constructive suggestion. We added the following reference (Jones PD 1987 Vaccine) on L163.

7) Figure 3e. The Y-axis is unclear – how did they determine 100% maximum binding? Is this based on antibody concentration or binding strengths, or both?

I apologize for the lack of clarity in the notation. The "percent binding to maximum" shown here is the ratio of the OD₄₅₀ value in the sample vs the OD₄₅₀ value which was given by IgG (anti-Narita-HA (1µg/ml), anti-PR8-HA (1µg/ml), anti-LAH (1µg/ml), rHA-2 (3µg/ml). Since the antibody concentrations are very, we cannot accurately quantify the results. However, these values still indicate the relative binding capacity of each clone. We revised Figure legends for **Fig.3e** (L764-L773).

8) Line 225, LAH needs reference

We add the reference of (Wang TT et al 2010 PloS Pathog) at L480.

9) Line 273, the conclusion that GC responses occur in the lung is not supported by their data, which show GC responses in the draining lymph node. The statement must be amended.

We agree that our indication was incorrect. Based on our data, the major GC response was observed in the MLN of the lung area (**Fig. 3a&b** and **Fig. 6d** and **S-Fig.7a**). Therefore, we corrected this description at L159-L171, L173-L190, L262-L269, L271-L272 and L347-L351.

10) Line 351, "Il-4 deficiency in B cells"....? This is not shown

This is a typo. "Il-4 deficiency" should be "IL-4R-deficient B cells". We corrected the sentence at L298-L299.

Reviewer #2 influenza vaccines, Tfh (Remarks to the Author):

The manuscript by Miyauchi et al explores the role of IL-4 in supporting germinal B cell function following infection or immunization. There is a tremendous amount of work presented, however the overall study suffers from a lack of clarity in the execution, analysis and rationale for the experiments carried out. While many of the models and techniques are technologically advanced, the limited explanation and justification for many of the experimental choices, combined with limited details about the reproducibility of the observations, make the strength of the authors observations very difficult to tease out. I suggest significantly reducing the scope of work presented to make the message more cogent.

We appreciate the reviewer's comments and have carefully revised the paper to improve the clarity in the rationale for the experiments and explanation. In the present manuscript, the following two points are the main focus.

1: The production of broadly reactive antibodies requires virus entry through the respiratory tract and viral replication in the deep lung, which induces a structural change in the virus HA that exposes rare antigenic epitopes. These rare epitopes are needed to activate a minor B cell population that produces broadly reactive antibodies.

2: As a result of virus infection, this minor B population differentiates and produces broadly reactive antibodies in the MLN. These B cells are amplified by the GC response, and IL-4 plays an important role in the efficient expansion of the B cells that respond to this rare antigen in the GC.

We summarized this issue in the abstract and the first paragraph (L321-L336) of discussion.

Major comments –

1 – For all experiments, the animals numbers per group and number of times each experiment was independently repeated is very unclear. For some panels, 2-3 animals are used while for others consist of groups of 7 or more. What is the justification for the different animal groups sizes? Similarly why are t-tests used for group comparison in some figures, but Mann whitney U tests used for later experiments. What is the

justification for selection of these tests and why change? Also, moving back and forth of the flow data from cell frequencies to absolute numbers presented in the column graphs is confusing, and total cell counts in each sample should be presented also in these instances.

We add several experiments to increase the number of animals and declared the animal numbers per group and the number of times for each experiment in Figure legends for **Fig.1 to 7** and **S-Fig.1 to 11**. We also revisited statistical analysis, using the Student's *t*-test for all data.

2 - Figure 1 shows cross-protection of mice against PR8 if pre-infected with pH1N1, but protection was not observed with prior immunization. Protection against H2N2 is not very convincing. Given the premise of this paper is that the quality of the antibody responses (specifically the degree of cross-reactive specificities) can confer heterologous protection, it is puzzling why the authors did not directly quantitate antibody against HA, NA, NP and the HA stem for both pdmH1N1 and PR8 (and H2N2) from their immunized and pre-infected animals.

We agree with the comments from this reviewer about the protection results of H2N2. The H2N2 strain had a serious problem in experimental reproducibility because its ability to infect mice was inconsistent. To resolve this problem, we establish a virus that could reproducibly infect mice. However, although it became a susceptible strain, the protection of the B6 mouse was no longer observed (**Fig. 1a**). Therefore, we revised the description of the neutralization breadth in the broad protection, noting that it was limited to H1N1 strains. We revised the sentence in L129-L133.

The reason why we did not directly measure the antibody titers against HA, NA, NP, and the HA stem is that serum from the infected mice is a mixture of single specific and dual specific antibodies, which cannot be perfectly distinguished by ELISA. In this manuscript, we focused on the broadly reactive antibody recognizing HA, which is a major target of neutralizing antibodies. In addition, since more than 70% of H1N1 is identical on an amino-acid base, it is not possible to determine the presence of the broadly reactive antibodies among the antibodies recognizing whole virus epitopes by ELISA. Therefore, *in vivo* measurement of neutralizing activity is the best available measure to prove the broad reactivity.

3 – Figure 1 - The serum and BALF transfer experiments are not that informative. These should be harvested from both vaccinated and pre-infected animals, and then the amount of antibody transferred equalized based on total amounts of anti-Narita binding Ig (or titres of anti-Narita antibody in serum at least). This would establish that qualitative differences in the antibody response, and not the magnitude of the responses underpin the differential protection observed against PR8 challenge.

We agree with the reviewer's comments. To more precisely quantify the neutralizing activity of the polyclonal antibodies obtained from the infected mice, we determined the input volume of serum or BALF based on the concentration of anti-HA antibodies in serum. We have re-examined the *in vivo* measurement of neutralizing activity after the normalization (**Fig.4c, 5a, and 6c**). We found no significant difference between before and after normalization. *Tmprss2*-, *Bcl6*- or *Ii4*-deficient mice consistently showed a reduction in anti-PR8 protection (70% of H1N1 is identical on amino-acid base). These results strongly support our notion that the production ability of the broadly reactive antibodies was largely dependent on the structural change occurred by virus replication, the GC response, and IL-4R signaling. We described the normalization strategy of serum in L233-L234, and Figure legends of **Fig.4c, 5a, and 6c**.

4 – Figure 1 – differential infusion volumes were used to control the depth of inocula (5ul for upper only, 50 for upper and lower). However, there was no concentration of virus in the mice administered 5ul, and the mice administered 50 showed consistently more virus in both upper and lower lung. The conclusion that lower lung is responsible for the antibody is not clear given more replication overall with the larger volume.

We apologize for the confusing description. As we mentioned in the response to the editor, this system was established to distinguish the infection site between the upper respiratory tract and the deep lung. This system allowed us to control the arrival point of the virus by changing the amount of liquid, which contains the same load of the virus. The justification of this setting is provided by measuring the virus load in the upper respiratory tract and the deep lung (**Fig.2b**). High virus load was found in the deep lung only when a larger volume of liquid was used for the infection. HA-specific antibodies were found in BALF only when high virus replication occurred in the deep lung. Therefore, we

concluded that heavy infection in the deep lung is essential for efficient antibody responses. We revised the explanation of this system in L150-L158.

5 – Figure 2 seems redundant. Basically mice that cannot sustain infection, don't generate antibodies, that then don't protect against challenge. Similarly to Sup Fig2. Note the WT controls offer very little protection in this experiment, although the dosing indicated is now given as pfu instead of LD50 in the first figure so is difficult to directly compare (and not indicated at all on Supp 2).

The significance of **Fig.2** is to show the relationship between viral replication and the antibody response. We showed that the sera from TMPRSS2-deficient infected mice protected from A/Narita/1/2009 infection but not from A/PR/8/1934 infection even after normalization of serum amounts was normalized by the anti-Narita-HA titer. Therefore, this Figure is important to show requirement of virus replication for generation of broadly protective antibodies. We prefer to keep this Figure (in revised version, **Fig.4**) and **S-Fig.2** (in revised version, **S-Fig.4**), and have revised an explanation for clarification in L218-L243, and fixed Figure legends for **S-Fig.4**. The notation of viral load has been standardized to LD₅₀.

6 - Figure 3A. Based on figure legend descriptors, the data shows splenic-derived B cell staining for the vaccinated animals via IP route. Do mice, instilled with inactivated virus via intranasal route, induce similar mono-specific Narita+ B cells in the MLN? There is a possibility that TMPRSS2 modifies the HA of the inactivated virus, despite the absence of replication, giving rise to cross-reactive epitopes. The data presented may indicate effects imparted by route of administration (IN vs IP) rather than the requisite for viral replication.

As this reviewer pointed out, we cannot rule out the possibility that the epitope required for broad-reactivity is created by TMPRSS2, but the virus replication cannot be completed in the absence of TMPRSS2. Therefore, currently, there is no available way to determine the relationship between TMPRSS2 and the epitopes required for broadly protective antibodies. However, it is generally accepted that the inactivated virus vaccines are including the HA structure after the conformational change. If this form is enough to induce broadly reactive antibodies, we expect to see the same antibodies even after vaccination.

However, vaccination consistently showed no broadly protective antibody (**Fig.1a & Fig.3a**); thus we consider this explanation unlikely.

We completely agree with the reviewer about the importance of the effects on the route of administration. To rule out the difference in the reactivity among different routes, the inactivated virus was administered intranasally to mice, but no antibody response was observed (see **additional data below**). Furthermore, to compensate for the insufficiency of the antigen dose in nasal administration of Tmprss2 deficient mice, we used more than 10,000 times higher virus load (3000LD₅₀) than in the regular setting (0.3LD₅₀). In this case, a protective antibody response against Narita was observed, but the broadly reactive antibodies were not detected (**Fig.4c, d**). These results were confirmed even after normalization with anti-Narita IgG titer. The antibody still fails to protect against the PR8 infection (**Fig.4c**). **Fig.3** also showed a clear difference in the breadth of the antibody responses between intraperitoneal and nasal administration. Therefore, these findings indicate that the difference in administration route cannot explain the difference in the breadth of the antibody responses. We added this explanation in the revised result on L218-L243.

7 – Figure 3 – It is unclear why the comparison is spleen (for vaccinated animals) and MLN (for infected), there could be anatomical differences that alter staining patterns of the HA probes. Both organs from both experimental conditions should be shown and analysed similarly by BCR sequencing and/or monoclonal generation. Also, it is unclear how many animals were analysed in figure 1a, please use dot plots instead of bar charts. Also, upstream flow cytometry gating trees need to be provided for each tissue.

In order to clarify the difference in the production ability of broadly reactive IgG antibodies between vaccination and infection, we examined the difference in the binding ability of B cells to the HA derived from Narita or PR8. As shown in **Fig.3a&b**, the broadly reactive IgG antibodies are not seen in the spleen during infection, although vaccination-induced single specific IgG was found in the spleen. On the other hand, vaccination is not enough to activate B cell responses in the lung dLN. Therefore, it is unlikely that the difference of immune response between the vaccination and the infection can be asked by the analysis in the same tissues. Under this circumstance, we believe that a comparison of B cell responses between the MLN in infection and the spleen in vaccination is the best available (**Fig.3a**). We added this explanation in the revised result on L159-L171, L173-L186, and L347-L355.

8 – Figure 3 - The BCR sequence analysis doesn't specify how many clones were recovered, how many animals were sorted? How variable was the repertoire between animals? How variable is the repertoire within an animal but between anatomical sites? Without this information, the data in fig3c and 3d and 3e don't have any meaning.

We apologize for the confusing description. In **Fig. 3 e&f**, we used the pooled B cells from the MLN of day 14 post-infection of two infected mice. A total of 720 clones were examined, 303 were IgG-producing B cells, and 111 clones provide readable V_H sequences. We had this explanation in L191-L216 of the result section.

9 – Figure 3 - How many mice were the Nojima culture antibodies derived from? Many studies have shown mice can make stem antibodies to infection or vaccination (Angeletti et al 2017, Tan et al 2019 Angeletti et al 2019) and can participate in GC reactions. The significance of the SHM data is unclear, especially given the very small number of clones analyzed. In either case, dual-specific or single-specific derived clones have very similar mutation loads. The authors assert that both “SHM of their immunoglobulin 241(Ig) genes is likely to be essential” and “SHM is not always necessary” for the generation of anti-stem antibodies. This is well established in the literature but not well supported by the included experiments in this study.

These are the data from the pooled MLN from two mice as we mentioned above. Our data is consistent with the previous reports mentioned by this reviewer, indicating the requirement of GC response for the production of stem antibodies. Analysis of the 49 clones that produced more than 50 µg/ml of IgG antibodies showed 19 clones recognizing the long alpha-helix (LAH) and the HA stalk domain. V_H regions of these clones did not have significant mutations (**Fig. 3f**). There was a subtle difference in the mutation frequencies between dual-specific and single-specific clones (**Fig. 3g**). Taken together, the experimental data in this study support the idea that “SHM is not always necessary” for the generation of anti-stem antibodies even in broadly reactive IgG antibodies. We revised the sentence in L191-L216 of the result section and L371-L387 of the discussion section.

10 – Figure 4 - BCL6 loss in T or B cells compromises the generation of a robust antibody response as expected. However, again there was no equalization of amounts of binding antibody in the BALF for the transfer experiment, so magnitude and quality changes in the serological response cannot be split apart.

We understand the point raised by the reviewer. To quantify the neutralizing activity more precisely, we determined the input volume of serum or BALF based on the concentration of anti-HA antibodies in serum (revised **S-Fig. 10**). We have re-examined the *in vivo* measurement of neutralizing activity after the normalization (revised **Fig.4c, 5a, and 6c**). We found no significant difference between before and after normalization. *Bcl6*-deficient mice consistently showed a reduction in anti-PR8 protection. These results strongly support our notion that the production ability of the broadly reactive antibodies was largely dependent on the GC response. We described the normalization of serum in L233-L234 and Figure legends of revised **Fig.4c, 5a, and 6c**.

11 – Figure 4 - The probe binding populations should be assessed on both the GC B cells (GL7+ or similar) and the bulk IgD⁻ population. Are the cross-reactive B cells in the GC or not?

Following this reviewer's suggestion, the content of dual-specific and single-specific B cells is shown in the GCB cells and the IgD⁻ population (revised **Fig.5c, and S-Fig.7c**). Approximately 70% of dual-specific B cells were GC B

cells, indicating the high GC dependency in the production of the broadly reactive antibodies. These results supported our idea that the dual reactive anti-HA antibody production in MLN is largely dependent on T_{FH} cell-regulated GC responses during acute infection (L257-L265).

12 – Figure 6 – It is interesting that IL-21 KO mice were able to maintain WT levels of GC activity. This seems contrary to the essential role IL-21 plays in GC formation, and implies that non-T cell sources of IL-21 can substitute for Tfh. Do the authors have an explanation for this?

This is a very important point. The role of the IL-21 in GC formation has been controversial. The normal GC development in *Il21*^{-/-} mice with NP-KLH immunization was reported by Leonard's group (Ozaki K and Leonard WJ et al 2002 Science). Tarlinton's group also observed normal GC formation in both *Il21*^{-/-} mice and *Il21*^{r/-} mice with NP-KLH immunization (Zotos et al 2018 JEM). Vinuesa's group reported that GC formation was abolished in *Il21*^{-/-} mice with SRBC immunization. The partial reduction of GC development in *Il21*^{-/-} mice with *T. gondii* infection and in *Il21*^{r/-} mice with the TLR stimulation, respectively, have been reported, (Stumhofer JS and Hunter CA et al 2013 PLoSOne, Bessa J and Bachmann MF et al 2010 JI). In our results using T cell-specific deficient mice, *Il21*^{fl/fl} Cd4-cre, GL-7^{high} GC-B cells, and T_{FH} cells were partially reduced in response to flu virus infection (revised **Fig.6d**). However, the BALF from deficient mice infected with the Narita strain showed adequate protection from PR8 infection (revised **Fig.6c**). We added this issue in L280-L294 of the result section.

13 – Figure 6. While the data shows anti-PR8 immunity is affected and the dual-probe population is reduced, the lack of protection may be rather a consequence of significant total GC B reduction. To show that the dual-probe population is indeed affected by IL4 deficiency, and not the narita+ GC B cells, experiments could be repeated with a Narita challenge model as well to show that IL4-deficiency still retains homologous protection.

Following this reviewer's suggestion, we reanalyzed the Narita infection model to understand the role of IL-4 in the protection against PR8 (revised **Fig.6c**). In order to more precisely quantify the neutralizing activity, we reanalyzed the input volume of serum based on the concentration of anti-HA antibodies in serum. As

we expected, IL-4 deficiency attenuated the protective response against PR8, but maintained the protective response against Narita. Similar results were found in *Il4ra*^{ΔB} mice (revised **Fig.6c**). These results demonstrate the importance of IL-4 and IL-4 signal in the GC response to PR8. To clarify, we revised in L285-L294 of the result section.

14 – Figure 7 - The change to OTII / OVA immunization models adds a further level of complication without addressing the initial premise. The authors should just use the bone marrow chimeras of the IL4Ra B cell KOs following Narita infection, you would expect only IL4Ra competent B cells to contain a dual probe population, while the KO B cells should show Narita probe staining only. That would help to address the total GC defect in the IL4Ra KO mice.

This is a very important experiment. As the reviewer's suggestion, we performed additional experiments using the Narita infection model with bone marrow chimeras of the IL-4R deficient and sufficient B cells. We found less proliferation of the IL-4Ra deficient B cells in the GC response (revised **Fig.7a**). We also find almost no dual specific B cells in the IL-4R deficient B cells (revised **Fig.7a**). We also found similar results in the IL-4R deficient B cells using OTII system, but we remove these data because of space limitation. These results strongly support our hypothesis that the IL-4 signal in the GCB cells has a critical role in the expansion of dual-specific B cells. We added and revised the sentence in L295-301.

Minor comments:

1 – “these human anti-stalk mAbs recognize conserved regions of the HA among group 1 and group 2 influenza A viruses, and the antibody heavy-chain variable region genes are heavily mutated.” - This is not supported in the literature and seem perfectly normal level of mutation for antigen experienced B cells.

This is a very important point. The role of the GC response on the broadly neutralizing antibodies is still controversial in the response to the HA antigen of the influenza virus. There are several pieces of evidence that the broadly neutralizing human antibodies are heavily mutated, suggesting the requirement of the GC response for the binding to common HA epitope (Lingwood D et al 2012 Nature, Pappas L et al 2014 Nature, Schmidt AG et al 2015 Cell). However,

there is not enough evidence to prove the requirement of the GC responses. In contrast, Lingwood group have demonstrated in a sophisticated way that pre-existing germline cording Ig can be a bnAb (Sangesland M et al 2019 Immunity). Our BCR sequence data support the idea that pre-existing B cells were partly responsible for the broadly neutralizing antibodies. Thus, we keep the sentence for this (L77-L81), and added the sentence “the germline cording human VH gene IGHV1-69 conferred pre-existing immunity by recognition of the bnAb epitope on the HA stalk without SHM” and reference (Sangesland M et al 2019 Immunity) on L83-L85.

2 - All flow plots should be moved off the axis, hard to clearly see the gated populations.

To clarify the dots of the gated populations, we switched to pseudocolor plots instead of contour plots for the FACS data (**Fig.3a, 4d, 5c, 6d,e, S-Fig.5, S-Fig.7a&c and Fig.11**).

3 – All column graphs should have individual data points.

The revised version provides individual data points in all of the figures.

Reviewer #3 antibody, vaccines (Remarks to the Author):

In this study, the authors performed a series of experiments on anti-influenza HA antibody responses to an inactivated vaccine or a wild type virus infection, focusing cross-reactive immunity using animal models (with mutant mice). The authors' major claims are that a live virus infection, through virus replication in the lung, induces more cross-reactive (broad) immunity which requires Tfh and germinal centre response, and IL-4 signalling is critical in the expansion of GC B cells recognising cross-reactive epitopes, thus important in activating broad anti-influenza (HA) antibody immunity.

Although it is already known a live infection would normally activates broader immunity and memory than an inactivated or subunit vaccine, this study provides important new insights on how a broader, cross-reactive antibody response to influenza (flu) HA is activated and demonstrates a critical role for Tfh-mediated IL-4 signalling in enhancing the broad anti-flu antibody responses targeting conserved HA epitopes and

immune memory. This may have implications in new vaccination strategy such as the development of "universal" flu vaccines.

Thank you for the positive comments.

1. On the implications from the study findings, it would be helpful to add a bit more discussion on how to use this knowledge to improve current vaccine or therapeutic strategy.

Thank you for helpful comments. According to the reviewer's advice, we added the discussion about therapeutic advantage at L415-L426 of conclusion paragraph.

2. On the use of different virus infection titres, i.e. 5ul or 50ul, to define upper or lower respiratory tract infection, is there any evidence or previous literature about this approach? No details were given on this aspect in the paper. In Figure 1e, it isn't clear to me the meanings of "Lung upper" and "Lung lower".

Also, it may be helpful to explain a bit more on how to determine where the antibodies are produced/generated, i.e. upper or lower respiratory tract.

This system was established to distinguish the infection site between the upper respiratory tract and the deep lung. This system allowed us to control a different arrival point of the virus by changing the amount of liquid, which contains the same titer of virus (0.3LD₅₀). The 5µl liquid is 10 times denser than the 50µl liquid. The justification of this setting is provided by measuring the virus titer in the upper respiratory tract and the deep lung (revised **Fig.2b**). High virus load was found in the deep lung only when the 50µl liquid was used for the infection. Therefore, we concluded that the antibody responses efficiently occur only when the virus infection is established in the deep lung. We revised the sentence of L150-L158.

We agree with "Lung upper" and "Lung lower" is misleading. We changed these to "Upper airway" and "Lung" at L150-L158, and Figure legend for revised **Fig.2b**.

To further answer the question of which secondary lymphoid organ is responsible for the antibody response during the infection, we added the ELISPOT analysis indicate that the MLN was the major site for induction of the

HA-specific antibody (**Fig.1e**). combined with **Fig.3a&b**, we concluded that the anti-HA IgG responses initially produced in the MLN associating with the lower respiratory tract and then become systemic (L168-L171).

3. Individual methods (methods section) seem rather brief, which may be limited by word limit.

We have revised the materials and methods section.

REVIEWERS' COMMENTS

Reviewer #1 (Remarks to the Author):

The authors have responded extensively and in full to previously raised concerns and suggestions.

Reviewer #2 (Remarks to the Author):

The resubmitted manuscript by Miyauchi et al explores the role of viral infection and IL-4 in supporting germinal B cell function following infection or immunization. The manuscript has been improved by narrowing the scope and adding additional experimentation. A few issues remain:

Major comments:

1 – line 85-86 – “In general, affinity maturation typically requires exceptional levels of SHM to efficiently generate bnAbs.”

This statement is not supportable, and is contradicted by the studies mentioned immediately prior that VH1-69 germline-derived bAbs are common in humans and carry very little mutation. This statement might be partially true for some classes of HIV-1 bNabs, but is in no way generalisable.

2 – regarding the inoculation volumes (5 vs 50ul). Again, no viral load was detected in mice given 5ul (in either the upper or lower tract), while 50ul instillation initiated infection at both locations. How can the authors conclude that lung infection is necessary, when only the larger dose drove any infection at either site? The authors state “HA-specific antibodies were found in BALF only when high virus replication occurred in the deep lung. Therefore, we concluded that heavy infection in the deep lung is essential for efficient antibody responses.” However this antibody could come from replication in the URT too. The text should be clarified.

3 – The authors now provide additional data that broadly reactive B cells are not evident in the spleen after infection. This further reinforces my original point that anatomical differences between the spleen and dLN responses need to be accounted for between their infection and immunisation systems. While I understand the authors use ip immunisation and hence favour the spleen, it should be straightforward to immunise im and show staining in an equivalent dLN to normalise the comparison to the infection data from the MLN. This should at a minimum be discussed and qualified further in the text.

Reviewer #3 (Remarks to the Author):

The study provides important insights on how a broader, cross-reactive antibody response to influenza HA is activated and demonstrates a critical role for Tfh-mediated IL-4 signalling in enhancing the broad antibody responses targeting the conserved HA epitopes and immune memory. The finding may have important implications for universal vaccination strategy. The authors appeared to have answered all the queries by the reviewers, and revised the manuscript accordingly which improved the clarity.

REVIEWERS' COMMENTS

Reviewer #1 (Remarks to the Author):

The authors have responded extensively and in full to previously raised concerns and suggestions.

We appreciate this reviewer and this comment.

Reviewer #2 (Remarks to the Author):

The resubmitted manuscript by Miyauchi et al explores the role of viral infection and IL-4 in supporting germinal B cell function following infection or immunization. The manuscript has been improved by narrowing the scope and adding additional experimentation. A few issues remain:

We appreciate this reviewer for careful reading and valuable comments.

Major comments:

1 - line 85-86 - "In general, affinity maturation typically requires exceptional levels of SHM to efficiently generate bnAbs."

This statement is not supportable, and is contradicted by the studies mentioned immediately prior that VH1-69 germline-derived bAbs are common in humans and carry very little mutation. This statement might be partially true for some classes of HIV-1 bNabs, but is in no way generalisable.

We agree with this reviewer. We removed this sentence as it is unnecessary (line 91).

2 - regarding the inoculation volumes (5 vs 50ul). Again, no viral load was detected in mice given 5ul (in either the upper or lower tract), while 50ul instillation initiated infection at both locations. How can the authors conclude that lung infection is necessary, when only the larger dose drove any infection at either site? The authors state "HA-specific antibodies were found in BALF only when high virus replication occurred in the deep lung. Therefore, we concluded that heavy infection in the deep lung is essential for efficient antibody responses." However this antibody

could come from replication in the URT too. The text should be clarified.

We agree with this reviewer. As this reviewer pointed out, the 50 μ l installation led to a high viral load in the upper and lower tracts. However, the infection site is not the main focus of this manuscript. Therefore, we revised the sentence as the following (line 157-159).

“The 50 μ l condition led to a high viral load in the upper and lower tract, indicating that this system effectively introduced the live virus into the deep-lung. Infection in the upper and lower tract resulted in higher anti-HA IgG levels in the BALF.”

3 - The authors now provide additional data that broadly reactive B cells are not evident in the spleen after infection. This further reinforces my original point that anatomical differences between the spleen and dLN responses need to be accounted for between their infection and immunisation systems. While I understand the authors use ip immunisation and hence favour the spleen, it should be straightforward to immunise im and show staining in an equivalent dLN to normalise the comparison to the infection data from the MLN. This should at a minimum be discussed and qualified further in the text.

According to the reviewer's suggestion, we applied intramuscular immunization, but we cannot see the enlargement of mediastinal LN. Moreover, we used intramuscular vaccination of inactivated Narita antigens using Alum or TiterMax adjuvants in the thigh muscle to see the response in popliteal LN. The popliteal LN showed Narita-specific B cells but not dual-specific B cells (Please see below data). Therefore, we concluded that local immune responses preferentially induced the broadly protective Abs during lung infection. We add a short sentence at line 355-357.

Reviewer #3 (Remarks to the Author):

The study provides important insights on how a broader, cross-reactive antibody response to influenza HA is activated and demonstrates a critical role for Tfh-mediated IL-4 signalling in enhancing the broad antibody responses targeting the conserved HA epitopes and immune memory. The finding may have important implications for universal vaccination strategy.

The authors appeared to have answered all the queries by the reviewers, and revised the manuscript accordingly which improved the clarity.

We appreciate this reviewer and this supportive comment.